# CO₂-evoked release of PGE2 modulates sighs and inspiration as demonstrated in brainstem organotypic culture

David Forsberg[1,2], Zachi Horn[1,2†], Evangelia Tserga[1,2†], Erik Smedler[3], Gilad Silberberg[4], Yuri Shvarev[1,2], Kai Kaila[5], Per Uhlén[3], Eric Herlenius[1,2*]

[1]Department of Women's and Children's Health, Karolinska Institutet, Stockholm, Sweden; [2]Karolinska University Hospital, Stockholm, Sweden; [3]Department of Medical Biochemistry and Biophysics, Karolinska Institutet, Stockholm, Sweden; [4]Department of Neuroscience, Karolinska Institutet, Stockholm, Sweden; [5]Department of Biosciences and Neuroscience Center, University of Helsinki, Helsinki, Finland

*For correspondence: eric. herlenius@ki.se

†These authors contributed equally to this work

**Abstract** Inflammation-induced release of prostaglandin E₂ (PGE₂) changes breathing patterns and the response to CO₂ levels. This may have fatal consequences in newborn babies and result in sudden infant death. To elucidate the underlying mechanisms, we present a novel breathing brainstem organotypic culture that generates rhythmic neural network and motor activity for 3 weeks. We show that increased CO₂ elicits a gap junction-dependent release of PGE₂. This alters neural network activity in the preBötzinger rhythm-generating complex and in the chemosensitive brainstem respiratory regions, thereby increasing sigh frequency and the depth of inspiration. We used mice lacking eicosanoid prostanoid 3 receptors (EP3R), breathing brainstem organotypic slices and optogenetic inhibition of EP3R$^{+/+}$ cells to demonstrate that the EP3R is important for the ventilatory response to hypercapnia. Our study identifies a novel pathway linking the inflammatory and respiratory systems, with implications for inspiration and sighs throughout life, and the ability to autoresuscitate when breathing fails.

## Introduction

Breathing is essential for life, but the underlying mechanisms that control breathing movements and neuronal pattern generation are under debate (*Jasinski et al., 2013*). Breathing maintains tissue homeostasis, and an adequate response to increased carbon dioxide (CO₂) levels is crucial (*Kaila and Ransom, 1998*; *Guyenet and Bayliss, 2015*). Failure to adequately respond to pCO₂ alterations is linked to breathing disturbances; apnea of prematurity; centrally mediated sickness, such as noxious sensations and panic; and premature death, as in sudden infant death syndrome (*Guyenet and Bayliss, 2015*).

Neuronal networks in the parafacial respiratory group/retrotrapezoid nucleus (pFRG/RTN) and the preBötzinger complex (preBötC) are important networks implicated in the central control of breathing. pFRG/RTN paired-like homeobox 2b (Phox2b)-expressing neurons are sensitive to changes in CO₂ levels or their proxy, pH ([H⁺]) (*Mellen and Thoby-Brisson, 2012*; *Onimaru and Dutschmann, 2012*). This responsiveness to hypercapnia is independent of synaptic transmission, and the Phox2b+ neurons detect CO₂/H⁺ via intrinsic proton receptors (TASK-2 and GPR4) in parallel pathways (*Kumar et al., 2015*). Moreover, medullary astrocytes contribute to central chemosensitivity. Slight acidification leads to an increased astrocytic intracellular concentration of calcium ions (Ca²⁺), resulting in vesicle-independent ATP release (*Gourine et al., 2010*).

**eLife digest** Humans and other mammals breathe air to absorb oxygen into the body and to remove carbon dioxide. We know that in a part of the brain called the brainstem, several regions work together to create breaths, but it is not clear precisely how this works. These regions adjust our breathing to the demands placed on the body by different activities, such as sleeping or exercising. Sometimes, especially in newborn babies, the brainstem's monitoring of oxygen and carbon dioxide does not work properly, which can lead to abnormal breathing and possibly death.

In the brain, cells called neurons form networks that can rapidly transfer information via electrical signals. Here, Forsberg et al. investigated the neural networks in the brainstem that generate and control breathing in mice. They used slices of mouse brainstem that had been kept alive in a dish in the laboratory. The slice contained an arrangement of neurons and supporting cells that allowed it to continue to produce patterns of electrical activity that are associated with breathing. Over a three-week period, Forsberg et al. monitored the activity of the cells and calculated how they were connected to each other. The experiments show that the neurons responsible for breathing were organized in a "small-world" network, in which the neurons are connected to each other directly or via small numbers of other neurons.

Further experiments tested how various factors affect the behavior of the network. For example, carbon dioxide triggered the release of a small molecule called prostaglandin E2 from cells. This molecule is known to play a role in inflammation and fever. However, in the carbon dioxide sensing region of the brainstem it acted as a signaling molecule that increased activity. Therefore, inflammation could interfere with the body's normal response to carbon dioxide and lead to potentially life-threatening breathing problems. Furthermore, prostaglandin E2 induced deeper breaths known as sighs, which may be vital for newborn babies to be able to take their first deep breaths of life. Future challenges include understanding how the brainstem neural networks generate breathing and translate this knowledge to improve the treatment of breathing difficulties in babies.

In addition, a $CO_2$ sensitivity of astrocytes also mediates a vesicular-independent ATP release (*Huckstepp and Dale, 2011*). Some connexins, which are expressed on astrocytes, e.g., connexin 26 (Cx26) and Cx30, are indeed sensitive to $CO_2$ (*Meigh et al., 2013*; *Reyes et al., 2014*).

These cellular processes of chemosensitivity result in an altered respiratory pattern that lowers the blood $CO_2$ levels. Inflammation reduces the $CO_2$ response and, particularly in neonatal mammals, can induce sighs, an altered response to hypoxia and potentially life-threatening apnea episodes as shown in humans, sheep, piglets and rodents (*Guerra et al., 1988*; *Long, 1988*; *Herlenius, 2011*; *Siljehav et al., 2014*; *Koch et al., 2015*; *Siljehav et al., 2015*).

In the inflammatory pathway, prostaglandin $E_2$ ($PGE_2$) is an important molecular mediator, that together with its main receptor, the EP3R, play roles in the hypoxic and hypercapnic responses, e.g. seen in patients with bronchopulmonary dysplasia (*Kovesi et al., 2006*; *Siljehav et al., 2014*; *Koch et al., 2015*). $PGE_2$ also seems to induce a sigh oriented respiratory pattern (*Koch et al., 2015*). Sighs are regularly occurring events of augmented breaths with a biphasic inspiratory pattern with the initial phase being comparable to eupnea and the second having larger amplitude (*Toporikova et al., 2015*). Such breaths are necessary for life and have been linked to several pathological states (*Ramirez, 2014*; *Li et al., 2016*).

Here, we hypothesized that both $PGE_2$ and EP3R constitute parts of the respiratory machinery and that they are involved in the induction of sighs and the hypercapnic response. We established a viable brainstem organotypic slice culture that maintains respiratory-related activity for several weeks in vitro and used this to investigate how $PGE_2$ and EP3R alter breathing and control of chemosensitivity. Our novel data reveal an important role of the EP3R in the pFRG/RTN hypercapnic response and furthermore suggest that $PGE_2$ is released during hypercapnia, possibly through $CO_2$-sensitive connexin hemichannels. Inflammation, with its associated $PGE_2$ release, exogenous $PGE_2$ and a lack of EP3R, blunts the hypercapnic response. These data link the inflammatory and respiratory systems,

with implications for sighs and inspiration throughout life as well as for the ability to autoresuscitate when breathing fails.

## Results

### EP3R is involved in respiratory control, sighs and the hypercapnic response

To investigate the role of $PGE_2$ and EP3R in respiration and sigh activity, we performed whole body plethysmography on 9-day old mice. We found EP3R and its ligand $PGE_2$ to be important modulators of breathing and the response to hypercapnia (5% $CO_2$ in normoxia; *Table 1*). The sigh frequency increased after the intracerebroventricular (i.c.v.) injection of $PGE_2$ (1 µM in 2–4 µl artificial cerebrospinal fluid, aCSF) in an EP3R-dependent manner (*Figure 1c–d*, *Table 2*), as did the tidal volume ($V_T$) (during eupnea, excluding sighs) in wild-type mice (*Figure 1e*). Furthermore, hypercapnic exposure also induced an increase in sigh frequency (*Figure 1f*, *Table 2*). This increase was larger in wild-type mice than in mice lacking the EP3R (*Ptger3-/-* mice). This $CO_2$-induced increase in sigh frequency was abolished in wild-type mice after i.c.v. injection of $PGE_2$ (*Figure 1f*, *Table 2*). The mice also responded to hypercapnia with increases in respiratory frequency ($F_R$), $V_T$ and minute ventilation ($V_E$; *Figure 1g*). I.c.v. injection of $PGE_2$ abolished the $V_T$ but not the $F_R$ response during hypercapnia (*Table 1*). This provides new information on how $PGE_2$ induces sigh activity and how increased PGE2 levels, as during inflammation, may both induce sighs and attenuate responsiveness to $CO_2$.

To unravel the mechanistic details of the $PGE_2$-EP3R system in respiratory regulation and its connection to the hypercapnic response and sighs, we set out to create a model system that would allow long-term, detailed studies of the respiratory neural networks, i.e., networks with neurons as well as glial cells.

### Establishment of a viable respiratory brainstem organotypic slice culture

Brainstem organotypic slice cultures of the mouse brainstem from 3-day-old mice were prepared at the preBötC brainstem level (*Figure 2a*). To validate this new model system, we first examined

**Table 1.** Respiratory parameters under basal conditions. *Ptger3-/-* mice are heavier than wild-type mice of the same age. They do not, however, differ in respiratory frequency ($F_R$), tidal volume ($V_T$), or minute ventilation ($V_E$). I.c.v. injection of $PGE_2$ increases $V_T$ and $V_E$ in wild-type mice but not *Ptger3-/-* mice. Respiratory frequency, tidal volume, and minute ventilation all increased during hypercapnic exposure. n: number of animals. Data are presented as mean ± SD.

| | Weight (g) | $F_R$ (breaths/min) | $V_T$ (µl /g) | $V_E$ (µl/g/min) | $F_R$ (breaths/min) Hypercapnia | $V_T$ (µl /g) Hypercapnia | $V_E$ (µl /g) Hypercapnia |
|---|---|---|---|---|---|---|---|
| WT - vehicle n=5 | 3.7 ± 0,5# | 206 ± 28* | 9.7 ± 2.9*† | 2.0 ± 0.8*† | 259 ± 20*# | 12.3 ± 3.1*# | 3.2 ± 1.0*# |
| WT - PGE2 n=8 | 3.9 ± 0.4 | 210 ± 15* | 15.1 ± 3.3† | 3.2 ± 0.8*† | 267 ± 31*# | 15.9 ± 2.6 | 4.2 ± 0.8*# |
| *Ptger3-/-* - vehicle n=5 | 4.8 ± 0.4# | 215 ± 32* | 14.2 ± 2.4* | 3.0 ± 0.5* | 240 ± 37*# | 15.7 ± 3.1*# | 3.4 ± 1.1*# |
| *Ptger3-/-* - PGE2 n=7 | 4.4 ± 0.3 | 211 ± 18* | 13.8 ± 2.7* | 2.9 ± 0.6* | 241 ± 30*# | 14.9 ± 2.7*# | 3.5 ± 0.4*# |

*p<0.05 (normocapnia vs. hypercapnia),

#p<0.05 (WT vs. *Ptger3-/-*),

†p<0.05 (vehicle vs. PGE2).

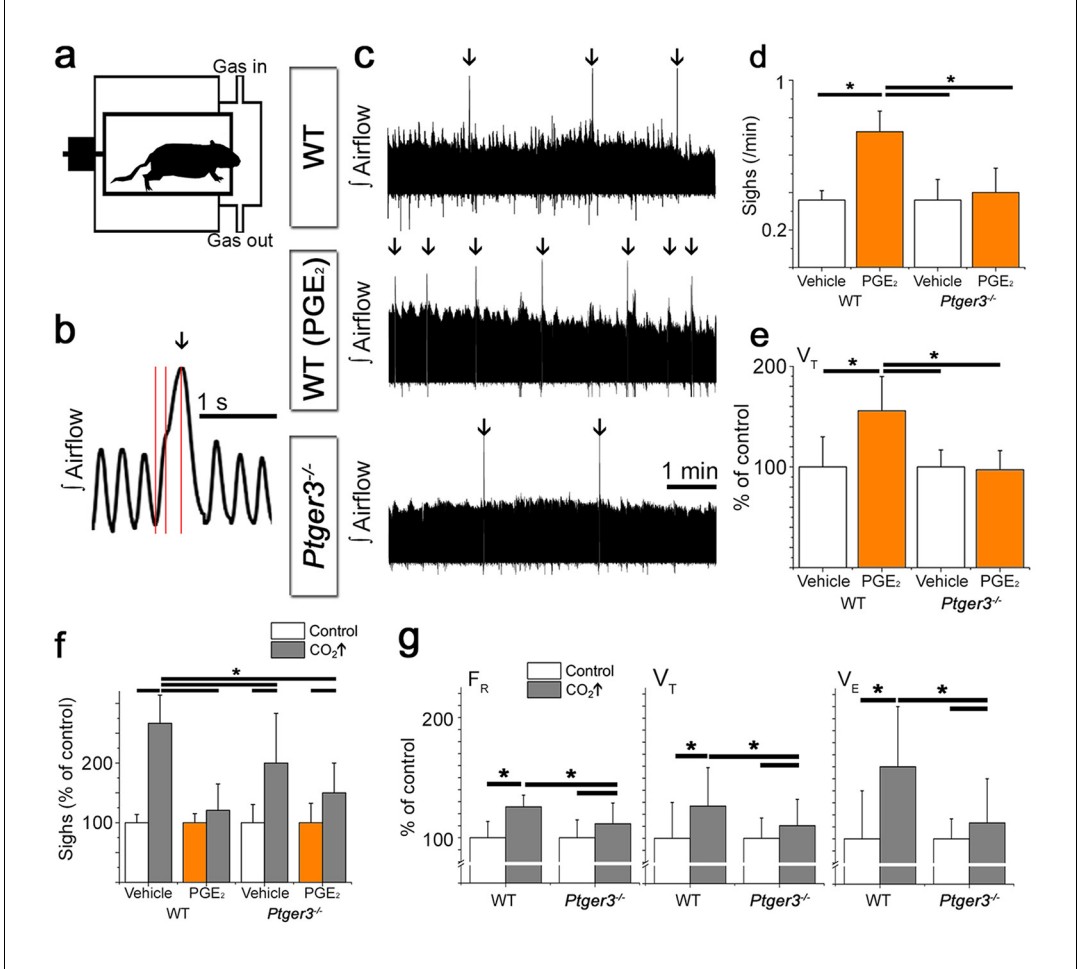

**Figure 1.** $PGE_2$ and $CO_2$ increase sigh activity via EP3R signaling. Respiratory activity was recorded in vivo in a two-chamber plethysmograph (a). Sighs, defined by an increase in inspiratory volume and respiratory cycle period with a biphasic inspiration (b), increase in frequency after intracerebroventricular injection (i.c.v.) of $PGE_2$. This effect is absent in mice lacking EP3R ($Ptger3^{-/-}$, c, arrows, d). I.c.v. injection of $PGE_2$ also increases the tidal volume ($V_T$) in wild-type C57BL/6J (WT) mice (e). The sigh frequency is increased by hypercapnic (5% $CO_2$ in normoxia) conditions in wild-type and $Ptger3^{-/-}$ mice but less so in $Ptger3^{-/-}$ mice (f). In wild-type mice, the increase is abolished after i.c.v. injection of $PGE_2$ (f). Hypercapnic exposure causes an increase in respiratory frequency ($F_R$), tidal volume ($V_T$), and minute ventilation ($V_E$) (g), but the increase is attenuated in $Ptger3^{-/-}$ mice. Data are presented as means ± SD. *$p<0.05$ Source data are available in a separate source data file.

The following source data is available for figure 1:

**Source data 1.** In vivo plethysmography data.

**Table 2.** $PGE_2$ and hypercapnia induce sighs. Sigh frequency does not differ between wild-type mice and $Ptger3^{-/-}$ mice. In wild-type mice, $PGE_2$ increases sigh frequency. Hypercapnia also increases sigh frequency more in wild-type mice than in $Ptger3^{-/-}$ mice. $PGE_2$ abolishes this increase in wild-type mice but not in $Ptger3^{-/-}$ mice (*$p<0.05$). n: number of animals. Data are presented as mean ± SD.

| | Sighs/min Normocapnia | Sighs/min Hypercapnia |
|---|---|---|
| WT - vehicle n=5 | 0.4 ± 0.1 | 1.0 ± 0.2* |
| WT - $PGE_2$ n=8 | 0.7 ± 0.1* | 0.9 ± 0.3 |
| $Ptger3^{-/-}$- vehicle n=5 | 0.4 ± 0.1 | 0.7 ± 0.3* |
| $Ptger3^{-/-}$— $PGE_2$ n=7 | 0.4 ± 0.1 | 0.6 ± 0.2* |

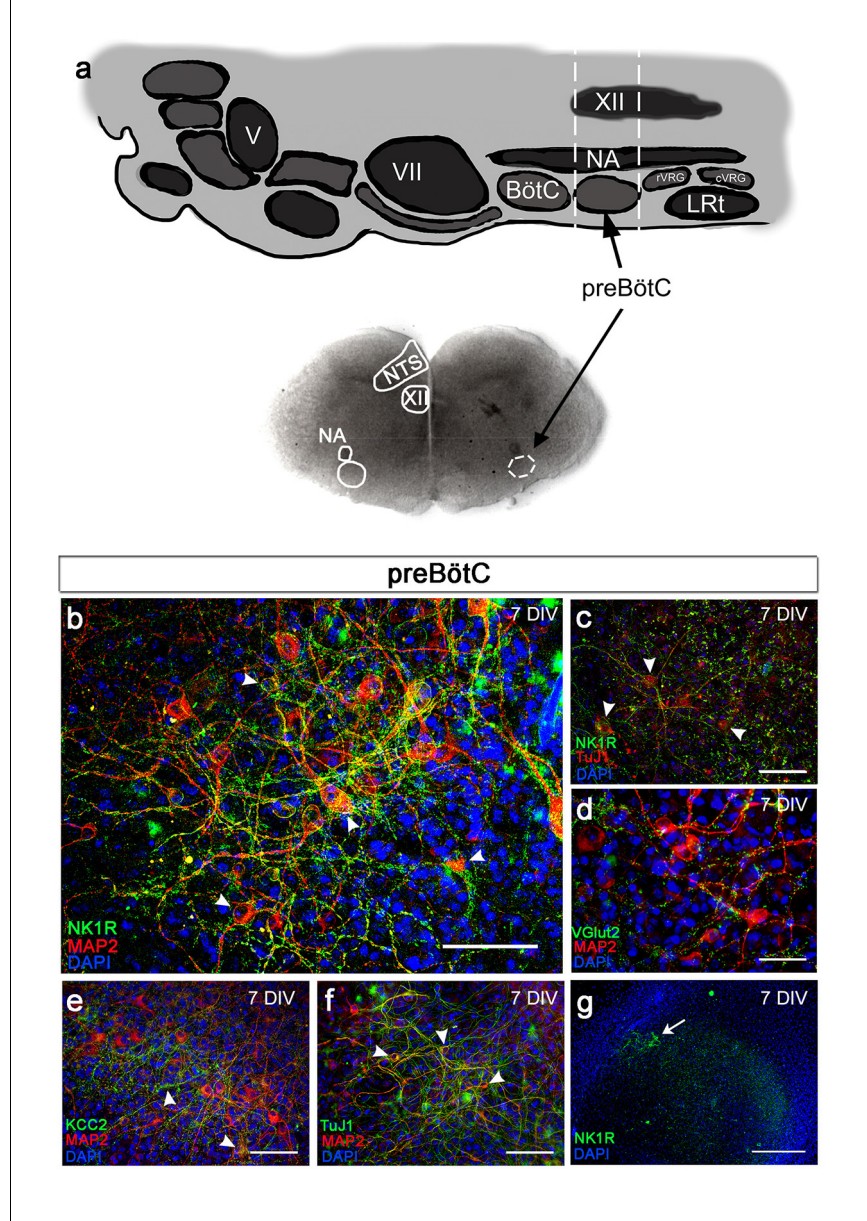

**Figure 2.** Brainstem slice cultures have a preserved structure and neurons with functional potential. Brainstem slices containing the preBötC were used to create slice cultures. Anatomical landmarks, including the nucleus ambiguus (NA), nucleus tractus solitarius (NTS), and nucleus hypoglossus (XII; **a**), as well as the distinct expression of NK1R (**b, c, g**) enabled the identification of the preBötC region. The brainstem slice displayed MAP2-/Tuj1-positive neurons expressing NK1R (**b, c**), VGlut2 (**d**), and/or KCC2 (**e**). The abundant MAP2-/Tuj1-positive cells demonstrated a preserved neuronal network within the preBötC (**g**). KCC2 expression was found in the NTS, NA, and preBötC (**e**). DIV; days in vitro. Arrowheads: double-labeled cells. Scale bars: 100 μm in **b–f**, 500 μm in **g**.

The following figure supplements are available for figure 2:

**Figure supplement 1.** Protein expression pattern is preserved during cultivation.

**Figure supplement 2.** Slices flatten during cultivation.

**Figure supplement 3.** Brainstem slice cultures are viable.

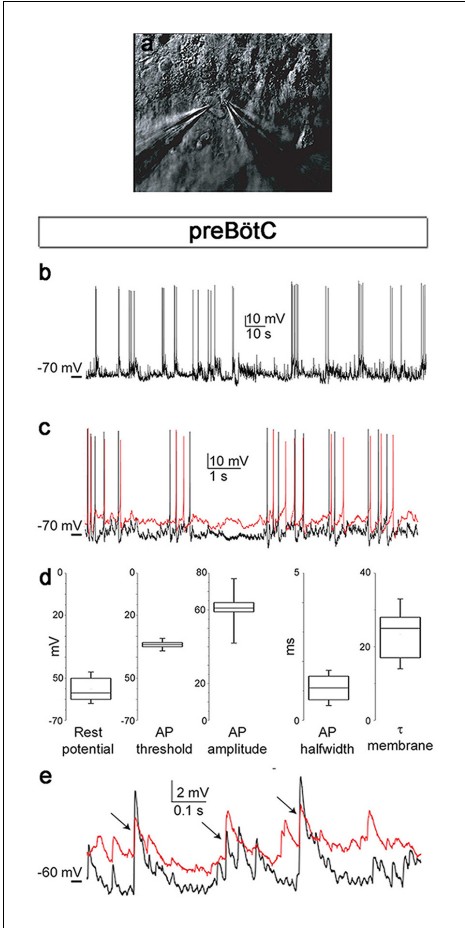

**Figure 3.** Neuronal electrical activity indicates preserved networks. Neurons in a preBötC slice (7 DIV), patched in the whole-cell configuration in current-clamp mode (a), exhibit regular rhythmic bursting activity (b). The neurons exhibited a hyperpolarized resting potential, action potentials, synaptic input, and spontaneous electrical activity, with epochs of action potential activity (b, c). The different measured variables indicated healthy and normally functioning neurons (d). Depicted here are two simultaneously patched neurons that also received common synaptic input (e, arrows). Spiking epochs occurred simultaneously, suggesting synchronized network oscillations. Direct connectivity between the depicted neurons showed that they were neither chemically nor electrically synaptically connected to each other. This finding indicates that the observed correlation was induced by common input from a preserved network structure. AP: action potential. DIV: days in vitro. Source data are available in a separate source data file.

The following source data and figure supplement are available for figure 3:

**Source data 1.** Electrophysiology patch clamp data.
**Figure supplement 1.** Cells of brainstem slice cultures retain neuronal electrical properties.

survival and expression of various neural markers in the brainstem slice cultures during cultivation.

Neural marker staining showed intact neurons, and neurokinin 1 receptor (NK1R)-positive respiratory regions were cytoarchitectonically well preserved (*Figure 2b,e,g*, *Figure 2—figure supplement 1*). The expression pattern of vesicular glutamate transporter 2 (VGlut2), similar to that in vivo, indicates the functional potential of the brainstem slice culture because glutamatergic synapses are essential for the development of the breathing rhythm generator (*Wallén-Mackenzie et al., 2006*) (*Figure 2d*). Neuronal markers MAP2 and KCC2 (*Kaila et al., 2014*) were expressed in the preBötC (*Figure 2c–f*, *Figure 2—figure supplement 2*). The protein expression in the preBötC remained stable for 3 weeks of cultivation (*Figure 2—figure supplement 1*). The brainstem slice cultures became thinner with longer cultivation as the tissue spread out (*Figure 2—figure supplement 2*). However, they remained viable and exhibited a low degree of necrosis and apoptosis, even after 3 weeks (*Figure 2—figure supplement 3*).

## Physiological measurements of brainstem respiratory activity demonstrate functional and responsive networks

After evaluating morphology, we investigated the cellular activity within the brainstem slice culture.

Neurons in the brainstem slice cultures retained their electrical properties at 7 days in vitro (DIV), including a resting membrane potential of $-55 \pm 6$ mV (*Figure 3b–c*) and overshooting action potentials (*Figure 3c*). The resting membrane potential, action potential threshold, half-width and peak amplitudes of the action potential, and membrane time constant were within the ranges of acute respiratory slices (*Figure 3c*, *Figure 3—figure supplement 1*). Action potentials occurred in clusters of regular rhythmic bursting activity. Neuronal connections were also similar to those seen immediately ex vivo, e.g., in acute slices, (*Ballanyi and Ruangkittisakul, 2009*) as evidenced by the postsynaptic potentials and concurrent inputs to neighboring neurons, resulting in correlated activity (*Figure 3b*, *Figure 3—figure supplement 1*).

Thus, on an individual neuronal level, the cells behave as expected. However, breathing is generated through cellular interactions in respiration-related neural networks.

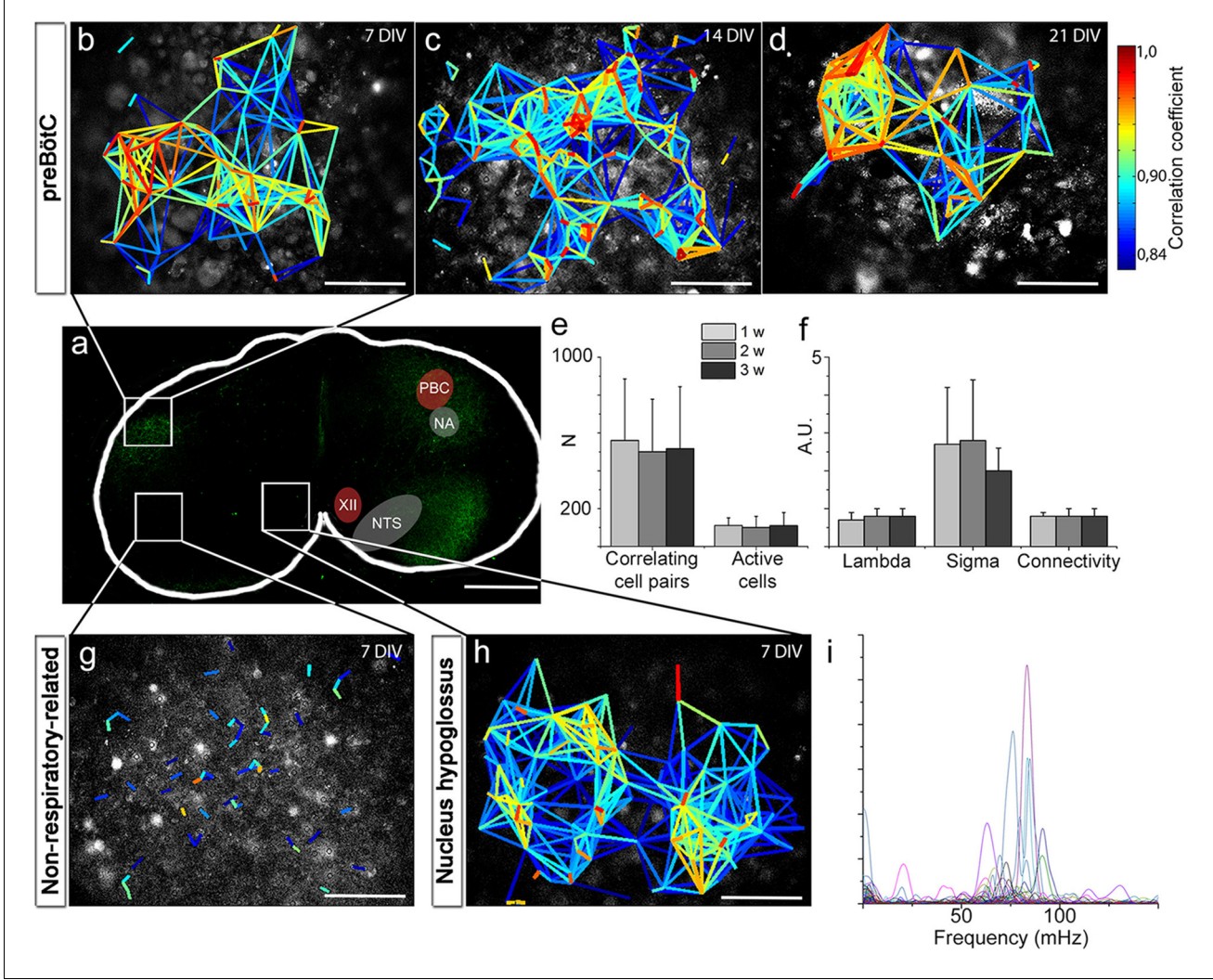

**Figure 4.** Neural activity in the preBötC is arranged in a functional respiratory network with respiratory-related motor output. In the preBötC slice (**a**), a cross-correlation analysis of Ca²⁺ time-lapse imaging data (*Figure 4—figure supplement 1*) revealed small-world network-structured correlated activity in the preBötC (**b–d**). The number of correlating cell pairs did not change over time (**e**), nor did the small-world network parameter or connectivity (**f**). TMR-SP-positive regions contained more correlated cell pairs than TMR-SP-negative regions (621 ± 284, N=14 and 56 ± 48, N=9, respectively; p<0.05), although there was no difference in the number of active cells (112 ± 57, N=14 and 144 ± 68, N=9, respectively, N.S.; **g**). As in the preBötC, the nucleus hypoglossus maintained correlated neural network activity (**h**). Ten percent of the cells (n=8–12/slice) in the hypoglossal nucleus exhibited a regular spiking frequency of ~50–100 mHz (**i**). The multicolored bar indicates the correlation coefficient in **b–h**; warmer colors indicate more strongly correlated activity between two cells connected by the line. DIV: days in vitro. A.U.: arbitrary units. w: week. N: number of slices, n: number of cells. Scale bars: 500 μm in **a**, 100 μm in **b–d** and **g–h**. Multicolored bar: color-coded correlation coefficient values. Data are presented as means ± SD. Source data are available in a separate source data file.

The following source data and figure supplements are available for figure 4:

**Source data 1.** Correlation data preBötC.
**Source data 2.** Frequency data with DAMGO.
**Figure supplement 1.** Single cell events provide information about correlated activity.
**Figure supplement 2.** Spontaneous Ca²⁺ activity is preserved for 3 weeks.

To investigate how individual cells interact, we applied live time-lapse Ca$^{2+}$ imaging to allow simultaneous recording of the activity of hundreds of cells. Tetramethyl rhodamine coupled Substance P (TMR-SP), visualizing NK1R-expressing neurons, was used to identify the preBötC. In the brainstem slice cultures, the preBötC contained networks with correlated activity between cells (*Figure 4b–d*), which was analyzed using a recently reported cross-correlation analysis method (*Smedler et al., 2014*) (*Figure 4—figure supplement 1*). We found clusters of cells with highly correlated activity. Such groups of cells in close proximity to each other were interconnected via a few cells that seem to function as hubs (*Watts and Strogatz, 1998*). The correlated network activity in the preBötC was preserved for 1, 2 and 3 weeks (*Figure 4b–e*). The number of active cells and the correlations per active cell remained similar over time (*Figure 4e*). These data suggest that the brainstem slice culture approach can indeed be used to perform long-term studies of respiratory neural network activity.

Analysis of the network structure revealed stable connectivity values (i.e., the number of cell pairs with a correlation coefficient exceeding the cut-off value, divided by the total number of cell pairs) during the cultivation of preBötC slices for up to 3 weeks (*Figure 4f*, *Table 3*). These values were slightly higher than those estimated in a previous study (*Hartelt et al., 2008*), in which only neurons were accounted for. However, both neurons and glia are involved in respiratory control (*Erlichman et al., 2010*; *Giaume et al., 2010*), and our analysis provides information on both cell types. Moreover, other analyzed network parameters, i.e., the normalized mean path-length ($\lambda$) and the normalized mean clustering-coefficient ($\sigma$), also remained stable (*Figure 4f*, *Table 3*). Overall, the small-world parameter (*Watts and Strogatz, 1998*) $\gamma = \frac{\sigma}{\lambda}$ was unchanged after 3 weeks in culture. Inhibiting the firing of action potentials and consequent activation of synapses by tetrodotoxin (TTX, 20 nM) abolished the coordinated network activity and revealed a population of cells that retained rhythmic alterations of cytosolic Ca$^{2+}$ levels ($31 \pm 4\%$ of the total number of cells, N=14 slices). Most of these cells ($76 \pm 12\%$, N=14) were NK1R-positive neurons, indicating the presence of functioning pacemaker neurons (*Figure 4—figure supplement 2*). The Ca$^{2+}$ signals from synapse-independent cells remained, however with a lower frequency and higher coefficient of variation (*Figure 4—figure supplement 2*). Regions outside the brainstem nuclei contained active cells, without intercellular coordination (*Figure 4g*). This cellular activity ceased during TTX treatment. In conclusion, the brainstem slice cultures contain a preserved preBötC network with a small-world structure.

As the preBötC delivers part of its motor output through the hypoglossal nerve (*Smith et al., 2009*), we also examined the hypoglossal motor nucleus. In this region of the hypoglossal motor nucleus, we found correlated cell activity organized similarly to that found in the preBötC network (*Figure 4h*). Within this network, frequency analysis revealed regularly spiking cells with a frequency between 50 and 100 mHz, corresponding to a rhythmic motor neuron output of 3–6 bursts of respiration-related activity/min (average $3.7 \pm 0.9$ bursts/min; *Figure 4i*). This suggests a preserved respiratory-related output in the brainstem slice cultures.

**Table 3.** The preBötC network parameters remain unchanged for 21-DIV cultures. The results of correlation analysis for the preBötC are shown. N.S.: not significant. N: number of slices. Data are presented as mean ± SD.

| preBötC | 7 DIV (N=12) | 14 DIV (N=13) | 21 DIV (N=8) | |
|---|---|---|---|---|
| Correlating cell pairs | 560 ± 325 | 501 ± 277 | 517 ± 327 | N.S. |
| Active cells | 110 ± 40 | 100 ± 59 | 110 ± 69 | N.S. |
| Correlations per active cell | 6 ± 4 | 6 ± 5 | 7 ± 6 | N.S. |
| Connectivity | 0.8 ± 0.1 | 0.8 ± 0.2 | 0.8 ± 0.2 | N.S. |
| Mean shortest path length (λ) | 0.7 ± 0.2 | 0.8 ± 0.2 | 0.8 ± 0.2 | N.S. |
| Clustering coefficient (σ) | 2.7 ± 1.5 | 2.8 ± 1.6 | 2 ± 0.6 | N.S. |
| Small-world parameter (γ) | 4.2 ± 3.0 | 3.4 ± 1.7 | 2.7 ± 1.7 | N.S. |

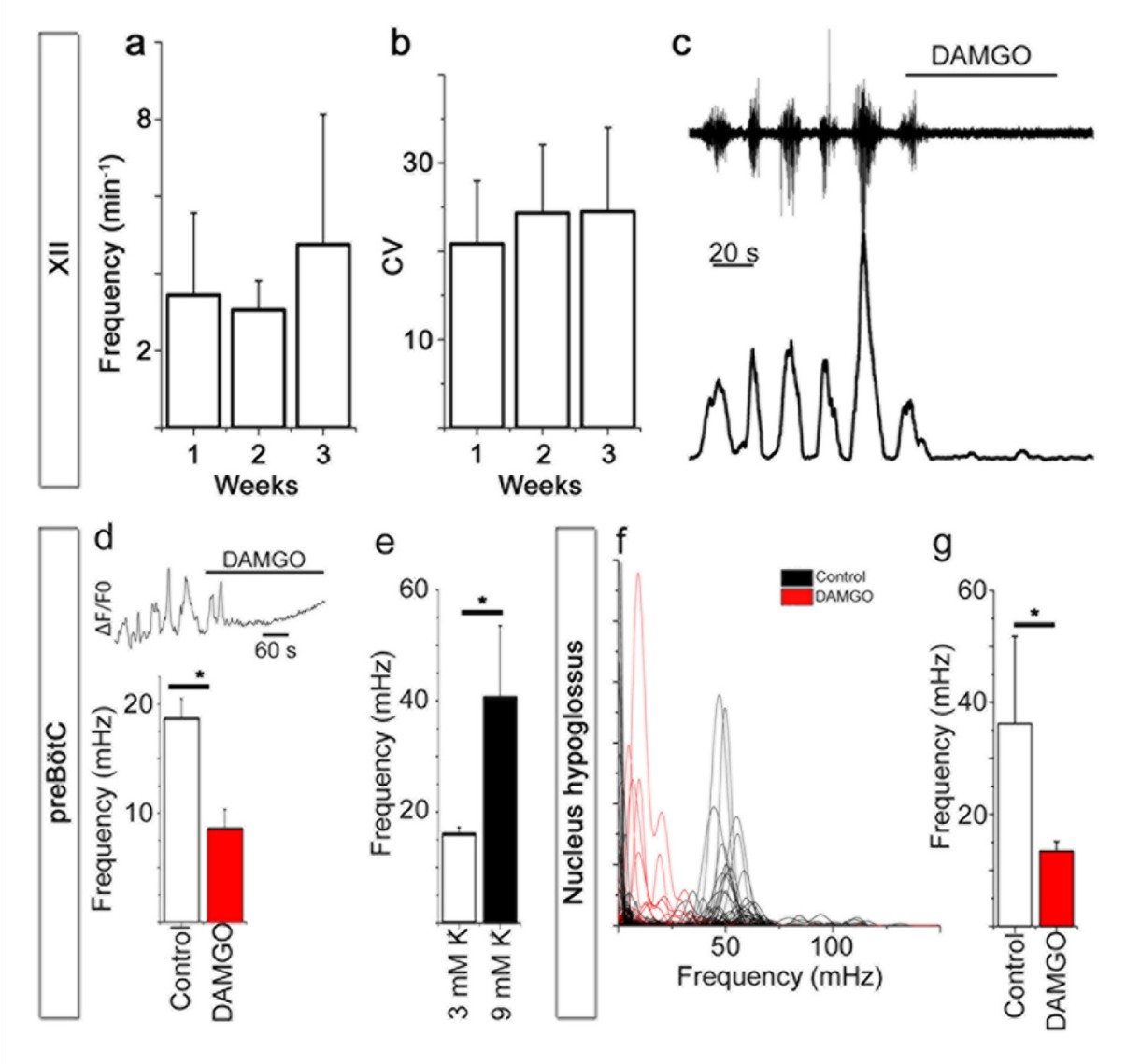

**Figure 5.** Breathing brainstem in a dish: ongoing/persistent rhythmic XII motor activity. The connected preBötC neural networks generate respiratory-related motor neuronal output delivered through the 12th cranial nerve (XII). The hypoglossal nucleus/nerve discharge frequency varied among the brainstem slice cultures but did not depend on brainstem slice culture age (a, N=16 at 7 DIV, N=3 at 14 DIV, and N=6 at 21 DIV). The regularity of respiration-related motor activity, measured as CV (coefficient of variation), remained stable during 3 weeks of culture (b). The μ-opioid receptor agonist DAMGO (0.5 μM) silenced the XII nerve activity in 5/5 brainstem slice cultures, as depicted here in (c) from a 7-DIV brainstem culture (filtered trace, above, and rectified and smoothed trace, below). DAMGO lowered the $Ca^{2+}$. In the hypoglossal nucleus, DAMGO (0.5 μM) lowered the frequency of regularly-spiking cells (f, g). N: number of slices. Data are presented as means ± SD. *p<0.05 Source data are available in a separate source data file.

The following source data and figure supplement are available for figure 5:

**Source data 1.** 12th cranial nerve electrophysiology recordings.
**Source data 2.** Frequency data with DAMGO.
**Source data 3.** High potassium frequency data.
**Source data 4.** Network topology and frequency data with DAMGO.
**Figure supplement 1.** Rhythmic respiratory-related output is preserved.

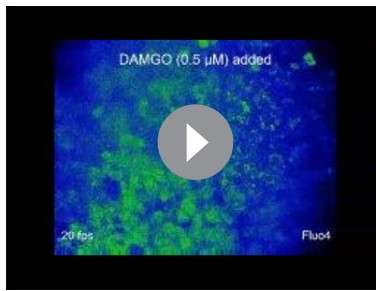

**Video 1.** NK1R[+] respiratory neurons in the preBötC are identified using TMR-SP (red dye), followed by Ca[2+] oscillations visualized with Fluo-4. After 25 s, the μ-opioid receptor agonist DAMGO (0.5 μM) is added and reduces the signaling frequency of the network. fps: frames per second.

Subsequent recordings of extracellular potentials from the 12[th] cranial nerve and hypoglossal nucleus revealed a corresponding rhythmic respiratory-related output at 7 (N=16), 14 (N=3), and 21 DIV (N=6). Respiratory output from acute slices varied between 1 and 8 bursts per min (neonatal mice, 3 mM K[+]), with frequencies in the lower range after a longer incubation time in vitro (*Ramirez et al., 1997*; *Ruangkittisakul et al., 2011*). In our model we observed a respiratory-related frequency of 3.7 ± 2.5 bursts per min (average of frequencies at 7, 14 and 21 DIV, no significant difference was observed between different DIV, *Figure 5a*), which is within the expected range for a slice. Among individual cultures, there was some variability in frequency (*Figure 5a*). However, the intrinsic rhythm was stable, with an average coefficient of variation of 22 ± 8 (no difference between the different DIV, *Figure 5b*). Rhythmic XII activity was observed for more than 2 hr during recordings (*Figure 5—figure supplement 1*).The activity could be inhibited by a μ-opioid receptor agonist, [D-Ala[2], N-Me-Phe[4], Gly[5]-ol]-enkephalin (DAMGO, 0.5 μM; *Figure 5c*, *Figure 5—figure supplement 1*) and stimulated by NK1R agonist Substance P (1 μM; 19 ± 13% increase in frequency, p<0.05; N=7; *Figure 5—figure supplement 1*).

In the preBötC, DAMGO also inhibited the Ca[2+] activity of individual NK1R[+] neurons and lowered the network frequency significantly (*Figure 5d*, *Video 1*). This was accompanied by an increase in the coefficient of variation in this area (36 ± 4 vs. 47 ± 6, N=7 slices, p<0.05). The network structure was not affected. An increase in [K[+]] from 3 mM to 9 mM, with subsequent membrane potential depolarization, increased the frequency in the preBötC (*Figure 5e*). In the hypoglossal nucleus, DAMGO caused a frequency reduction in the regularly spiking cells (*Figure 5f,g*). Thus, the preBötC brainstem slice culture remained active and responsive and generated rhythmic respiration-related motor output activity.

## Gap junctions are essential parts of correlated preBötC activity

Gap junction signaling plays an important role in the development of the respiratory system, the maintenance of respiratory output and likely the CO₂/pH response (*Elsen et al., 2008*; *Fortin and Thoby-Brisson, 2009*; *Gourine et al., 2010*; *Huckstepp et al., 2010a*). Thus, we used the brainstem slice cultures to investigate the involvement of gap junctions in the neural networks and their response to CO₂.

In the brainstem slice cultures, immunohistochemistry showed high Cx43 expression in neurons of the preBötC (*Figure 6a*) and lower and persistent Cx26 and Cx32 expression in the respiratory regions (*Figure 6b–d*) at 7 DIV. To assess the function of these intercellular gap junctions and hemichannels, we treated the brainstem slice cultures at 7 DIV with gap junction inhibitors carbenoxolone (CBX) or 18-α-glycyrrhetinic acid (18-α-GA). Both inhibitors decreased the number of correlating cell pairs and active cells in the preBötC, whereas glycyrrhizic acid (GZA), an analog to CBX that lacks the ability to block gap junctions, and the aCSF control did not (*Figure 6e–g, k–l*). However, the individual activity of NK1R expressing neurons was not affected (*Figure 6h–j, m*). These findings suggest a role for gap junctions in the maintenance of correlated network activity in the preBötC.

Conversely the rhythmic activity of NK1R[+] neurons does not depend on gap junctions. Moreover, gap junction inhibition did not affect the mean correlation values, connectivity, or small-world parameter of the remaining correlated cell pairs (*Figure 6—figure supplement 1*). This demonstrates that the cells connected in a gap junction-independent manner are organized as a small-world network. These results are in line with topological data showing that respiratory neurons are organized in small clusters in the preBötC (*Hartelt et al., 2008*).

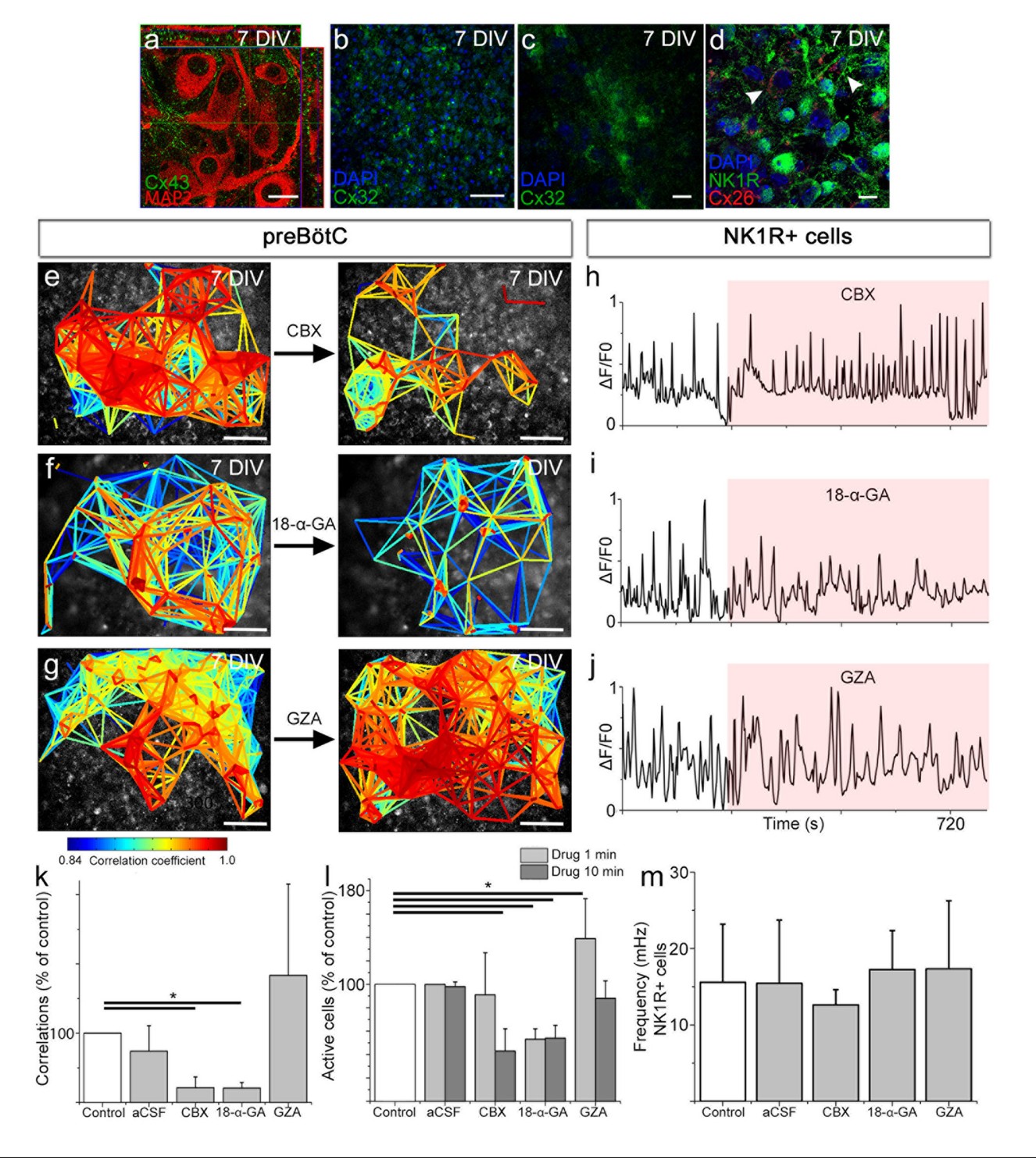

**Figure 6.** Gap junctions are necessary to maintain part of the correlated respiratory network. In the respiratory regions, the gap junction proteins Cx43 (**a**, N=9), Cx32 (**b**, **c**, N=8), and Cx26 (**d**, arrowheads; double-labeling with NK1R, N=5) are present. Gap junction inhibitors CBX (**e**) and 18-α-GA (**f**) reduced network synchronization in the preBötC. Notably, the $Ca^{2+}$ activity of individual NK1R-positive cells was not affected (**h–j**, **m**). Correlating cell pair numbers decreased to 21% (N=8) and 20% (N=6) of their respective controls after treatment with CBX and 18-α-GA, respectively (**k**). Network properties were not affected by GZA, an analog to CBX that lacks the ability to block gap junctions, (**g**, **j–k**, N=7) or aCSF (N=8). An initial increase in fluorescence intensity was noted after adding CBX and GZA but not after adding 18-α-GA, indicating an immediate excitatory effect of CBX and GZA (**l**). 18-α-GA reduced the number of active cells in the network at 1 min after application (53%), but CBX did not (91%, N.S.). At the same time point, an increased number of active cells were observed with GZA treatment (139%). After 10 min, a reduction of the number of active cells was found after

*Figure 6 continued on next page*

*Figure 6 continued*

treatment with both 18-α-GA and CBX (54% and 43%). However, the number of active cells returned to normal after GZA application (89%, N.S.; I). DIV: days in vitro. N: number of slices. Scale bars: 10 μm in a, c, and d, 100 μm in others. Multicolored bar: color-coded correlation coefficient values. Data are presented as means ± SD. *p<0.05. Source data are available in a separate source data file.

The following source data and figure supplement are available for figure 6:

**Source data 1.** Gap junction inhibition data.

**Figure supplement 1.** A gap junction-independent network is present within the preBötC.

## PGE$_2$ modulates preBötC activity

Our in vivo data, as well as others', indicate that PGE$_2$ and hypercapnia induce sigh activity (*Ramirez, 2014*; *Koch et al., 2015*). We hypothesized that this is due to effects on the respiratory centers in the brainstem. We used our brainstem slice cultures of the preBötC to study the direct effects of PGE$_2$ and hypercapnia in vitro.

PGE$_2$ levels in cerebrospinal fluid measured in experimental models and in human infants are in the pico- to nanomolar range (*Hofstetter et al., 2007*). In the brainstem slice cultures at 7 DIV, the application of PGE$_2$ (10 nM) lowered the Ca$^{2+}$ signaling frequency of respiratory neurons in the preBötC (*Figure 7a–b*). PGE$_2$ also induced longer Ca$^{2+}$ transients, and the signal amplitudes increased compared to those of the controls (*Figure 7b*). Koch and colleagues (*Koch et al., 2015*) suggested that the increase in sighs induced by PGE$_2$ is mediated through persistent sodium channels (I$_{NaP}$) (*Koch et al., 2015*). Indeed, in the preBötC, 10 μM Riluzole, a blocker of the persistent sodium current (I$_{NaP}$), attenuated effect of PGE$_2$ on Ca$^{2+}$ signal amplitude and length as well as decreasing the signal frequency (*Figure 7b*). As in previous studies (*Toporikova et al., 2015*), Riluzole did not affect the Ca$^{2+}$ signal compared to control periods. Riluzole is used as an I$_{NaP}$ blocker, but may also affect other parts of neuronal signaling, such as glutamate release (*Wang et al., 2004*). Therefore, we cannot completely determine whether the PGE$_2$ effect is due to an effect on the persistent sodium current or interference with glutamate signaling, although an effect on I$_{NaP}$ is likely (*Koch et al., 2015*).

EP3Rs were present in the preBötC (*Figure 7c–d*). qRT-PCR showed that 20% of the EP3Rs were of the α-subtype (*Figure 7e*). EP3Rα inhibits adenylate cyclase via Gi-protein, and reduced cAMP levels inhibit F$_R$ (*Ballanyi et al., 1997*). The EP3Rγ subtype, however, which couples to the G$_S$-protein, was the most abundant (*Figure 7e*).

In vivo, hypercapnia increases sigh activity, V$_T$, F$_R$, and V$_E$ (*Figure 1*). Therefore, we exposed the preBötC brainstem slice culture to increased levels of CO$_2$ by raising the pCO$_2$ levels from 4.6 kPa to hypercapnic 6.6 kPa, while maintaining a constant pH of 7.5 in the aCSF by the addition of bicarbonate. This did not have any effect on the Ca$^{2+}$ signaling frequency, the Ca$^{2+}$ signaling pattern or the network structure in wild-type or *Ptger3$^{-/-}$* mice (*Figure 7f–g*, *Figure 7—figure supplement 1*). However, the preBötC is not the main central chemosensitive region. Instead, the sensitivity to CO$_2$ is more profound in the pFRG. Therefore, we generated organotypic slice cultures of the pFRG/RTN brainstem level.

## The pFRG/RTN respiratory region exhibited correlated network activity and retained CO$_2$ sensitivity

The analysis of network structure and function that we conducted on the preBötC was previously not possible to perform in the pFRG/RTN on acute transverse slices. Studies of the pFRG/RTN are particularly interesting because of its crucial role in central respiratory chemosensitivity (*Onimaru et al., 2009*). We therefore created the same type of brainstem slice culture as with the preBötC slice using slices containing the pFRG/RTN instead (*Figure 8a*). These brainstem slice cultures expressed neuronal markers as expected (*Figure 8b–d*, *Figure 8—figure supplement 1*) and displayed retention of electrical properties, in a manner similar to the preBötC brainstem slice cultures (*Figure 8e–f*).

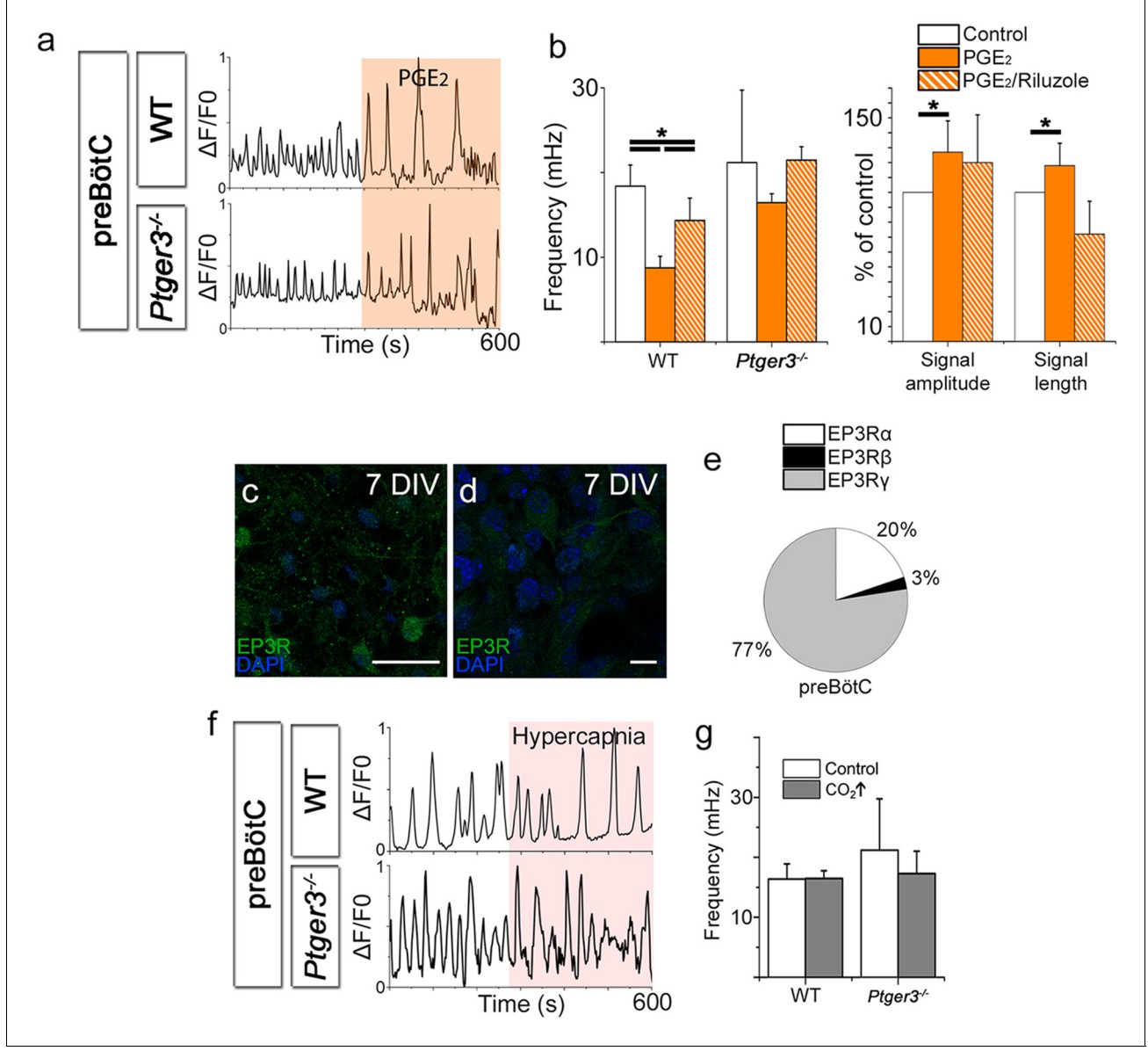

**Figure 7.** $PGE_2$ modulates preBötC network activity. $PGE_2$ lowered the $Ca^{2+}$ signaling frequency of the preBötC network in WT mice but not in *Ptger3*[-/-] mice (**a–b**). The effect was attenuated but not abolished by Riluzole (**b**). $PGE_2$ also increased signal amplitude and length (**a–b**), an effect that was abolished after Riluzole application (**b**). *Ptger3* is expressed in the preBötC (**c, d**), and 20% of the EP3Rs were of the α ($G_i$-protein coupled) subtype and 77% of the γ ($G_s$-protein coupled) subtype (**e**). Hypercapnic exposure ($pCO_2$ elevated from 4.6 to 6.6 kPa) did not affect the signal frequency of the preBötC (**f–g**). DIV: days in vitro. Scale bars: 50 μm in c and 10 μm in **d**. *$p < 0.05$ Source data are available in a separate source data file.

The following source data and figure supplement are available for figure 7:

**Source data 1.** PGE2 data preBötC.
**Source data 2.** Hypercapnia data preBötC.
**Source data 3.** Hypercapnia data preBötC 2.
**Figure supplement 1.** Hypercapnia had no effect on the preBötC.

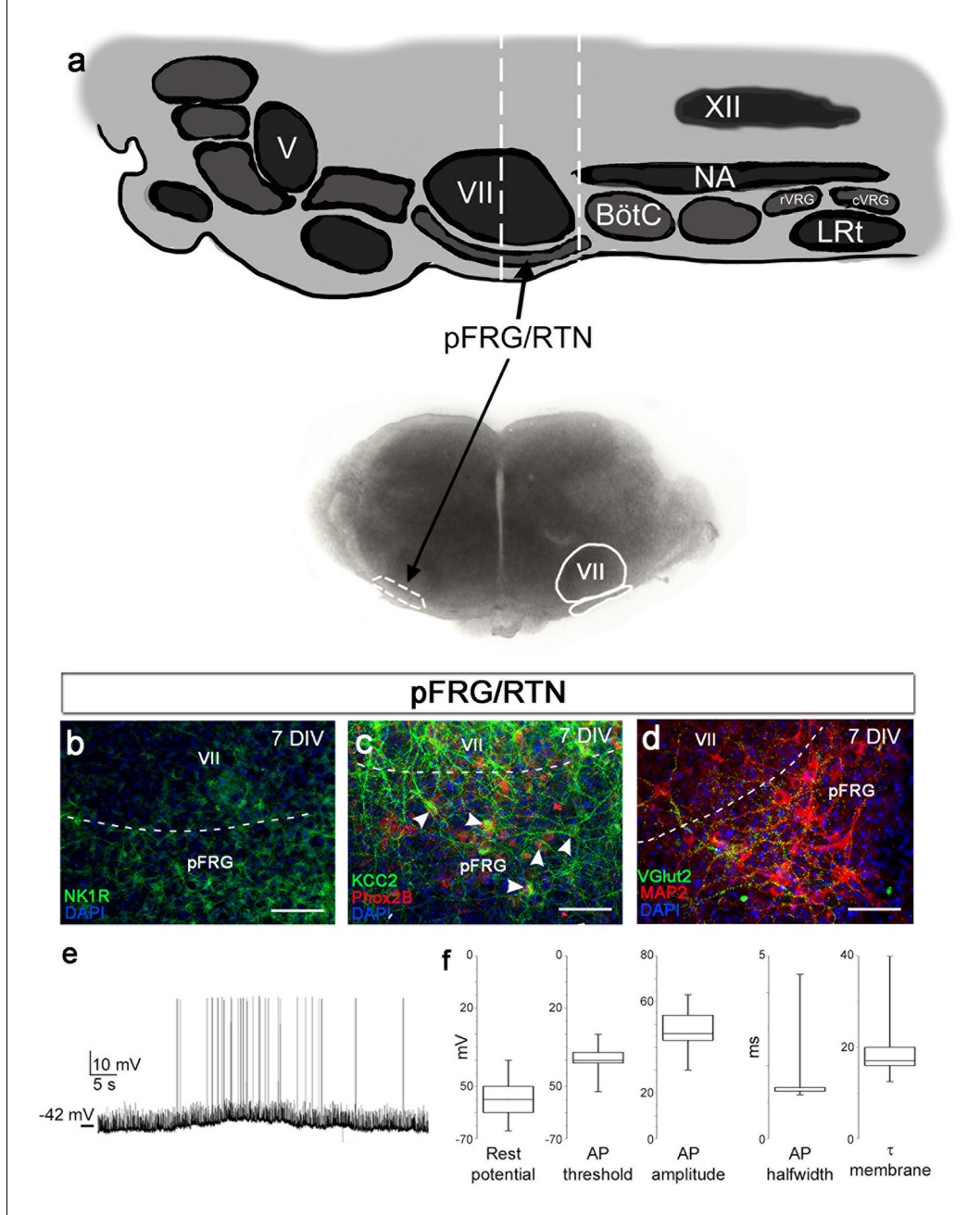

**Figure 8.** pFRG/RTN brainstem slice culture. pFRG/RTN slices were selected based on the location of the facial nucleus (VII; **a**). In the brainstem slice culture, pFRG/RTN expressed the neuronal markers NK1R (**b**), KCC2 (**c**), Phox2b (**c**), vGlut2 (**d**), and MAP2 (**d**). The pFRG/RTN neurons also retained adequate electrical properties and generated spontaneous action potentials individually or in clusters (**e–f**). Data are presented as box plots with minimum and maximum values. DIV: days in vitro. Scale bars: 100 µm.

The following source data and figure supplement are available for figure 8:

**Source data 1.** pFRG/RTN characterization.

**Figure supplement 1.** Cultivation of pFRG/RTN slices.

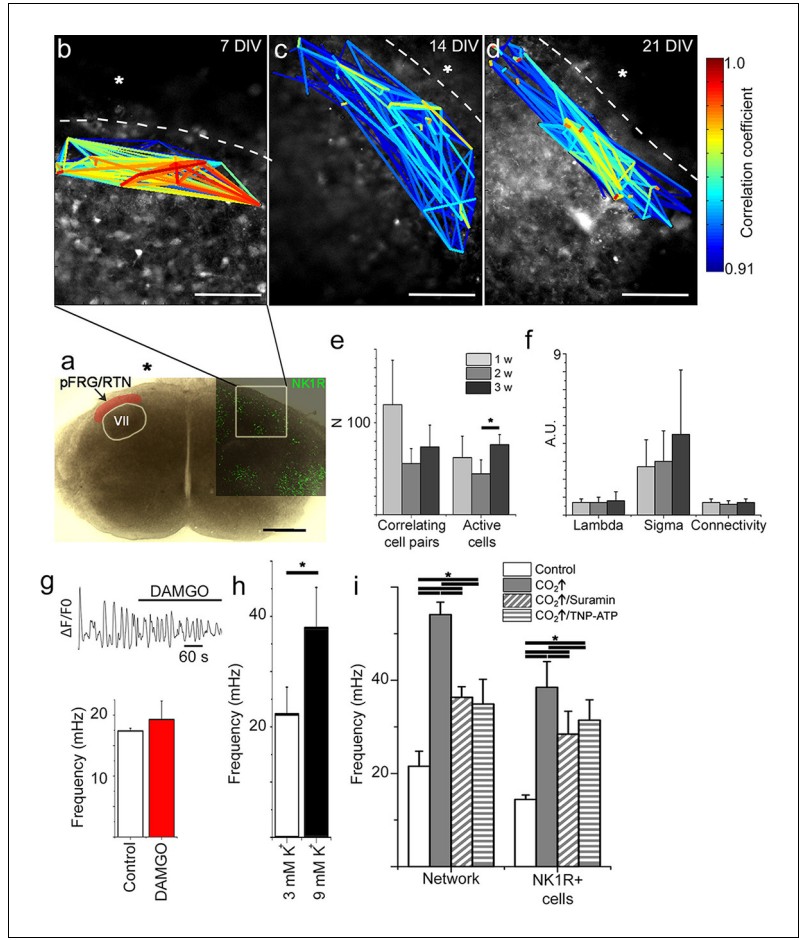

**Figure 9.** The pFRG/RTN respiration-related network generates correlated neural activity and responds to $CO_2$. The pFRG/RTN network is arranged in a small-world manner just ventral to the facial nucleus. The network structure was preserved during cultivation (**a–d**). The number of correlating cell pairs did not change with longer culturing times, but the number of active cells was higher at 3 weeks than at 2 weeks ($45 \pm 27 < 76 \pm 19$, $p < 0,05$; **e**). The network parameters were stable during cultivation (**f**). The pFRG/RTN network did not respond to the μ-opioid receptor agonist DAMGO (0.5 μM; n=420, N=4; **a**), but the average network frequency increased with higher potassium concentrations ($22 \pm 5$ mHz and $38 \pm 7$ mHz, N=12; **b**). Both the neural network and individual NK1R/TMR-SP-labeled cells responded to increases in $CO_2$ pressure ($pCO_2$ elevated to 6.6 kPa), indicating that the chemosensitivity was preserved in the pFRG/RTN brainstem slice culture. Suramin, a P2 receptor antagonist, and TNP-APT, a P2X receptor antagonist, attenuated the $CO_2$ response but did not abolish it (**g**). DIV: days in vitro. Scale bars: 100 μm. Multicolored bar: color-coded correlation coefficient values. N: number of slices, n: number of cells. Data are presented as means ± SD. *p<0.05. Source data are available in a separate source data file.

The following source data and figure supplements are available for figure 9:

**Source data 1.** Correlation data pFRG/RTN.

**Source data 2.** Hypercapnia data.

**Source data 3.** High potassium frequency data.

**Source data 4.** Riluzole and TTX data.

**Figure supplement 1.** Spontaneous $Ca^{2+}$ activity is preserved during cultivation.

**Figure supplement 2.** Hypercapnia reduces mean path lengths in the pFRG/RTN of wild-type mice.

**Table 4.** The pFRG/RTN network parameters remain unchanged for 21-DIV cultures. The results of correlation analysis for the pFRG/RTN are shown. Among the analyzed network parameters, only the number of active cells differed at the analyzed time points, and only between 14 and 21 DIV. N.S.: not significant. N: number of slices. Data are presented as mean ± SD.

| pFRG/RTN | 7 DIV (N=12) | 14 DIV (N=11) | 21 DIV (N=6) | |
|---|---|---|---|---|
| Correlating cell pairs | 118 ± 69 | 61 ± 31 | 74 ± 42 | N.S. |
| Active cells | 49 ± 26 | 41 ± 21* | 76 ± 19* | *p<0.05 |
| Correlations per active cell | 3.1 ± 2.2 | 1.7 ± 1.1 | 1.0 ± 0.7 | N.S. |
| Connectivity | 0.7 ± 0.2 | 0.6 ± 0.2 | 0.7 ± 0.2 | N.S. |
| Mean shortest path length ($\lambda$) | 0.7 ± 0.2 | 0.7 ± 0.3 | 0.8 ± 0.5 | N.S. |
| Clustering coefficient ($\sigma$) | 2.7 ± 1.6 | 3.0 ± 1.7 | 4.5 ± 3.6 | N.S. |
| Small-world parameter ($\gamma$) | 3.6 ± 2.5 | 4.2 ± 2.6 | 3.3 ± 1.6 | N.S |

Looking at multiple cells using time-lapse $Ca^{2+}$ imaging, the activity of the pFRG/RTN was correlated in a scale-free small-world network, akin the one in the preBötC (*Figure 9b–d*) and was stable during cultivation (*Figure 9e–f*). There was a slight difference in the number of active cells between 2 week and 3 week cultures (*Figure 7e*). However, all network properties remained unchanged (*Figure 9f* and *Table 4*). The inhibition of neuronal spiking and synapses by TTX (20 nM) disrupted the coordinated activity (21 ± 9% of correlated cell pairs remained, N=11). However, rhythmic $Ca^{2+}$ activity persisted in a subset of primarily (64 ± 9%, N=11) NK1R-positive cells (*Figure 9—figure supplement 1*). The pFRG/RTN cells did not exhibit any change in signaling frequency after DAMGO application (*Figure 9g*, average levels from 7-, 14-, and 21-DIV cultures are displayed, as there were no significant differences among cultures of these ages), confirming the absence of preBötC μ-opioid-sensitive regions in these slices (*Ballanyi and Ruangkittisakul, 2009*). Similarly to the preBötC brainstem slice culture, the pFRG/RTN responded to higher $[K^+]$ with an increase in frequency (*Figure 9h*; average levels from 7-, 14-, and 21-DIV cultures are displayed, as there were no significant differences among cultures of these ages).

Next we examined the $CO_2$ sensitivity of the pFRG/RTN (*Onimaru et al., 2008*). This resulted in increased signal frequency of the $Ca^{2+}$ oscillations (*Figure 9i*, *Table 5*, *Video 2*; data from 7-DIV cultures are displayed, and no significant differences in the response among 7-, 14-, and 21-DIV cultures were observed) and the activation of some previously dormant cells. During hypercapnic

**Table 5.** pFRG/RTN slices respond to $CO_2$ if the EP3R is present. The average mean frequency of all cells in the network and the average mean frequency of NK1R-positive cells during the control period or during exposure to hypercapnia are shown ($pCO_2$ = 55 mmHg, pH = 7.5). N.S.: not significant. N: number of slices, n: number of cells. Data are presented as mean ± SD.

| | Mean frequency (mHz) | | | | | |
|---|---|---|---|---|---|---|
| | **Network** | | | **NK1R+ cells** | | |
| | **Control** | **Hypercapnia** | | **Control** | **Hypercapnia** | |
| pFRG/RTN - WT (N=7, n=343) | 21.6 ± 3.2 | 54.1 ± 2.7* | p<0.05 | 14.4 ± 0.9 | 38.5 ± 5.5* | p<0.05 |
| pFRG/RTN - *Ptger3*$^{-/-}$ (N=5, n=448) | 25.0 ± 7.9 | 26.0 ± 1.9 | N.S. | 11.4 ± 5.8 | 11.6 ± 3.8 | N.S. |
| preBötC - WT (N=5, n=1737) | 16.4 ± 2.5 | 16.5 ± 1.3 | N.S. | 16.6 ± 4.6 | 15.7 ± 5.3 | N.S. |
| preBötC - *Ptger3*$^{-/-}$ (N=4, n=822) | 21.1 ± 8.6 | 17.3 ± 3.8 | N.S. | 22.7 ± 5.9 | 17.8 ± 7.7 | N.S. |

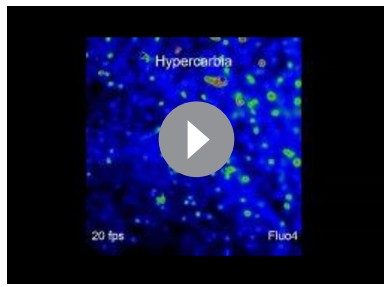

**Video 2.** Ca$^{2+}$ oscillations visualized with Fluo-4 in the chemosensitive region pFRG/RTN. Low network activity is increased by exposure to hypercapnia after 15 s. fps: frames per second.

exposure, the pFRG/RTN network topology remained essentially unchanged (*Figure 9—figure supplement 2*).

Response to hypercapnia involves pFRG/RTN astrocytes, which release ATP that acts on purinergic P2-receptors (*Erlichman et al., 2010*; *Gourine et al., 2010*; *Huckstepp et al., 2010a*). We sought to examine whether this kind of signaling pathway was active in the 7-DIV brainstem slice cultures, and we found that blocking purinergic receptors with Suramin or TNP-ATP application did not abolish the hypercapnic response, in agreement with previous data (*Sobrinho et al., 2014*). However, both the unspecific P2 receptor and the more specific P2X receptor antagonist attenuated the $CO_2$ response by approximately one third (30 ± 6%; *Figure 9i*), as observed in adult and neonatal rats (*Wenker et al., 2012*) and 9-day-old mice (*Gourine et al., 2010*). Thus, the $CO_2$-induced release of ATP acting on P2 receptors may contribute to the $CO_2$ response.

In conclusion, our brainstem organotypic slice culture contains an active pFRG/RTN network that retains its structural integrity over time and responds to $CO_2$ exposure with increased activity.

## The $CO_2$ response is dependent on EP3R signaling and gap junctions

Gap junctions, both intercellular and hemichannels, are linked to respiratory chemosensitivity (*Huckstepp et al., 2010a*; *Meigh et al., 2013*; *Reyes et al., 2014*). Recently, $CO_2$ was shown to interact with the hemichannel Cx26, inducing an open state through the formation of carbamate bridges, thus increasing the release of compounds such as ATP (*Meigh et al., 2013*). Therefore, we hypothesized that gap junctions exert functions within the pFRG/RTN network. However, gap junction inhibitors did not affect signaling frequency or network topology of the pFRG/RTN (*Figure 10a*, *Figure 10—figure supplement 1*). Instead, the frequency response to hypercapnia was both inhibited and reversed by the application of the gap junction inhibitor 18-α-GA (*Figure 10b–c*). GZA (a structural analog of CBX without gap junction-inhibiting properties) did not alter the $CO_2$ response (*Figure 10b–c*).

We conclude that 18-α-GA inhibits the hypercapnic response, while inhibition of purinergic signaling pathways attenuates it. Thus, we suggest that the $CO_2$ response is not entirely explained by the connexin-mediated release of ATP. Furthermore, inflammation via $PGE_2$ and EP3R alters the hypercapnic response in vivo and in brainstem spinal cord *en bloc* preparations (*Figure 1* and Siljehav and colleagues Figures 1 and 4 [*Siljehav et al., 2014*]). Therefore, we hypothesized that hypercapnic responses involve $PGE_2$ signaling and next analyzed the $PGE_2$ content of the aCSF under control and hypercapnic conditions. In all examined slices (N=12/12, 7 DIV), a transient doubling of the $PGE_2$ concentration after $pCO_2$ elevation was evident (*Figure 11*). When gap junction blockers were applied, this peak was absent (N=4/4, 7 DIV; *Figure 11*). This indicates a hypercapnia-induced, gap junction-mediated release of $PGE_2$.

Immunohistochemistry showed expression of microsomal prostaglandin E synthase 1 (mPGEs-1) in GFAP positive astrocytes (*Figure 11—figure supplement 1*). mPGEs-1, the main $PGE_2$ producing enzyme, has previously been found mainly in endothelial cells of the blood brain barrier of adult rats (*Yamagata et al., 2001*). Our findings suggest that astrocytes in the vicinity of the ventral brainstem border of neonates express mPGEs-1 and might therefore be candidates for modulation of breathing through $CO_2$-induced release of $PGE_2$.

$PGE_2$ has a primarily inhibitory effect on respiration in neonatal mice and humans (*Hofstetter et al., 2007*), which we confirmed to account for its effects on the preBötC (*Figure 7*). However, as hypercapnia seems to induce a release of $PGE_2$ while stimulating breathing activity, we hypothesized that $PGE_2$ has a direct stimulatory effect on the pFRG/RTN. Indeed, $PGE_2$ increased the signaling frequency of pFRG/RTN neurons (*Figure 12a–b*, *Table 6*). This effect was

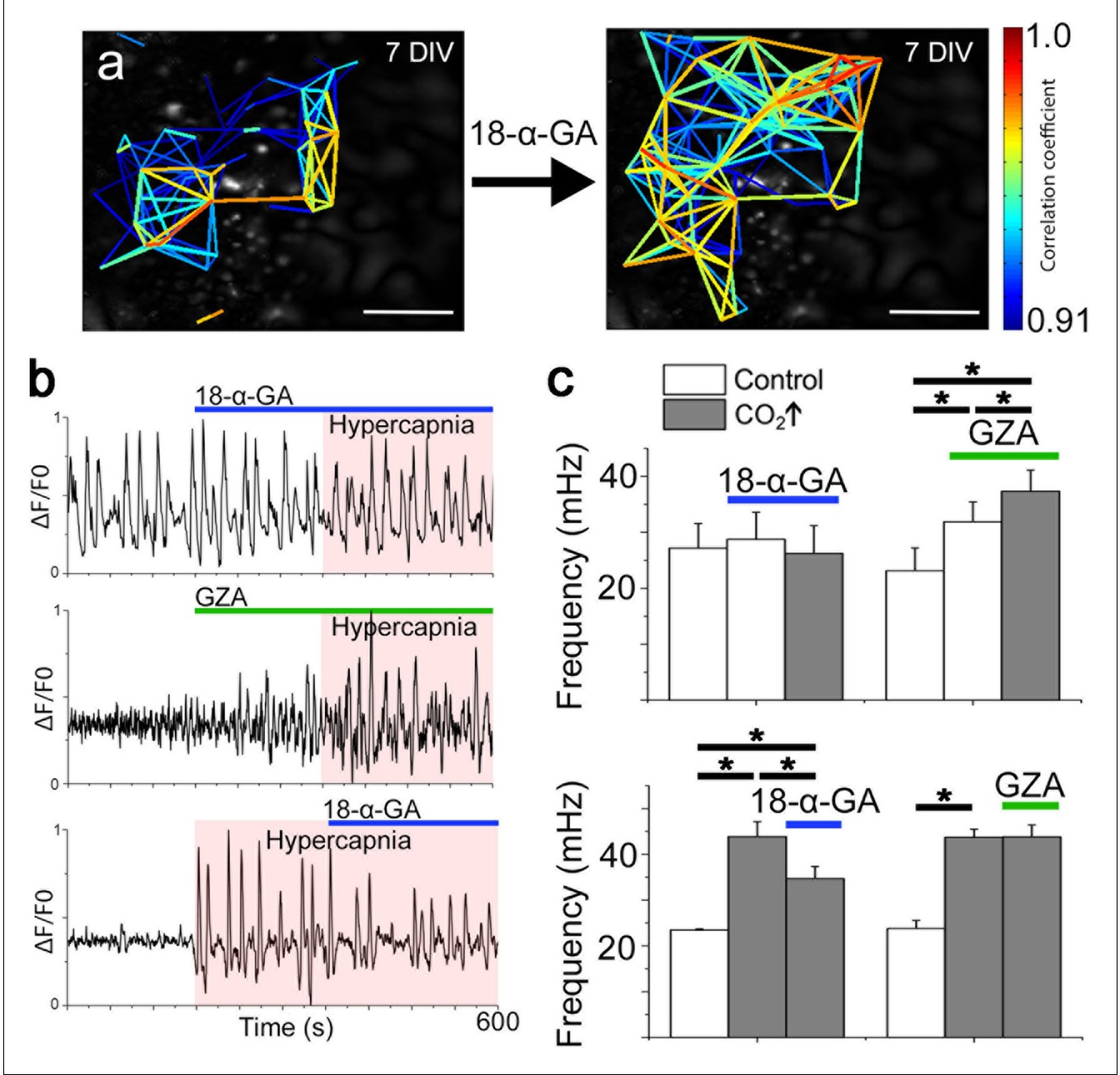

**Figure 10.** Correlated pFRG/RTN network activity is not dependent on gap junctions, but hypercapnic responses are. Blocking gap junctions in the pFRG/RTN did not change the functional network structure of the respiratory center or alter its frequency (a and c, N=7). However, hypercapnic responses ($CO_2\uparrow$) were abolished when gap junctions were inhibited by 18-α-GA (b, top trace; c, left graph, N=7). GZA (a structural analog of CBX without gap junction-inhibiting properties) increased the frequency, and hypercapnia increased it further (b, middle trace; c, middle graph, N=7). An initiated hypercapnic response was attenuated but not completely reversed by 18-α-GA (b, bottom trace; c, lower graph, N=5). This dynamic was not seen after application of GZA. DIV: days in vitro. Scale bars: 200 μm. N: number of slices. Multicolored bar: color-coded correlation coefficient values. Data are presented as means ± SD. *$p<0.05$. Source data are available in a separate source data file.

The following source data and figure supplement are available for figure 10:

**Source data 1.** Hypercapnia and gap junction inhibition frequency data.

**Source data 2.** Hypercapnia and gap junction inhibition network data.

**Figure supplement 1.** Network structure in the pFRG/RTN is not dependent on gap junctions.

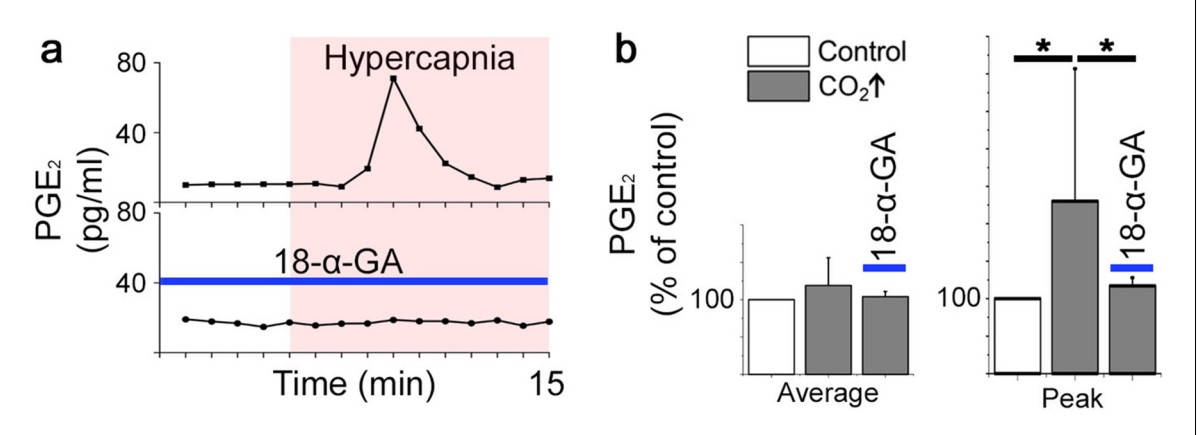

**Figure 11.** PGE$_2$ is released during hypercapnia. The aCSF contents exhibited an increase in microenvironmental PGE$_2$ levels during hypercapnia in 12 out of 12 slices. Here, the PGE$_2$ concentration of a brainstem slice culture is displayed during control and hypercapnic periods (**a**). When gap junctions were inhibited (18-α-GA, blue line), the PGE$_2$ levels remained unaltered during hypercapnia (N=4). The average PGE$_2$ level throughout the whole experiment was not affected by hypercapnia, but the peak value was higher during hypercapnia than under control conditions (**b**). N: number of slices. Data are presented as means ± SD. *p<0.05. Source data are available in a separate source data file.

The following source data and figure supplement are available for figure 11:

**Source data 1.** Hypercapnia PGE$_2$ ELISA data.
**Figure supplement 1.** mPGEs-1 is expressed in astrocytes in the proximity of the ventral border of the pFRG.

EP3R dependent, and EP3Rs were present in the pFRG/RTN, expressed both on respiratory neurons and on astrocytes (*Figure 12c–e*). We also observed a non-significant increase in amplitude (8 ± 3% and 11 ± 4% increase compared to control period, N.S.). Neither the PGE$_2$ effect nor the hypercapnic response of the pFRG/RTN was affected by Riluzole (30 ± 5 mHz vs 25 ± 2 mHz, N.S., N=6, and 36 ± 2 mHz vs 35 ± 6 mHz, N.S., N=6). qRT-PCR showed abundant expression of the EP3Rγ subtype, which couples to the G$_S$-protein (*Namba et al., 1993*). This would lead to an increase in intracellular cAMP in the pFRG/RTN *Ptger3*-expressing cells in response to PGE$_2$ (*Figure 12f*).

**Table 6.** PGE$_2$ increases the frequency of pFRG/RTN neurons and decreases the frequency of preBötC neurons. The mean frequencies of NK1R-positive cells during the control period or during exposure to 10 nM PGE$_2$ are shown. N.S.: not significant. N: number of slices, n: number of cells. Data are presented as mean ± SD.

| | Mean frequency (mHz) | | |
| --- | --- | --- | --- |
| | Control | PGE$_2$ | |
| pFRG/RTN - WT (N=5, n=343) | 13.7 ± 1.1 | 21.5 ± 2.9* | p<0.05 |
| pFRG/RTN - *Ptger3−/−* (N=4, n=448) | 12.1 ± 2.0 | 8.5 ± 2.9 | N.S. |
| preBötC - WT (N=7, n=1737) | 20.3 ± 2.2 | 8.7 ± 1.4* | p<0.05 |
| preBötC - *Ptger3−/−* (N=5, n=822) | 22.8 ± 2.3 | 16.4 ± 1.1 | N.S. |

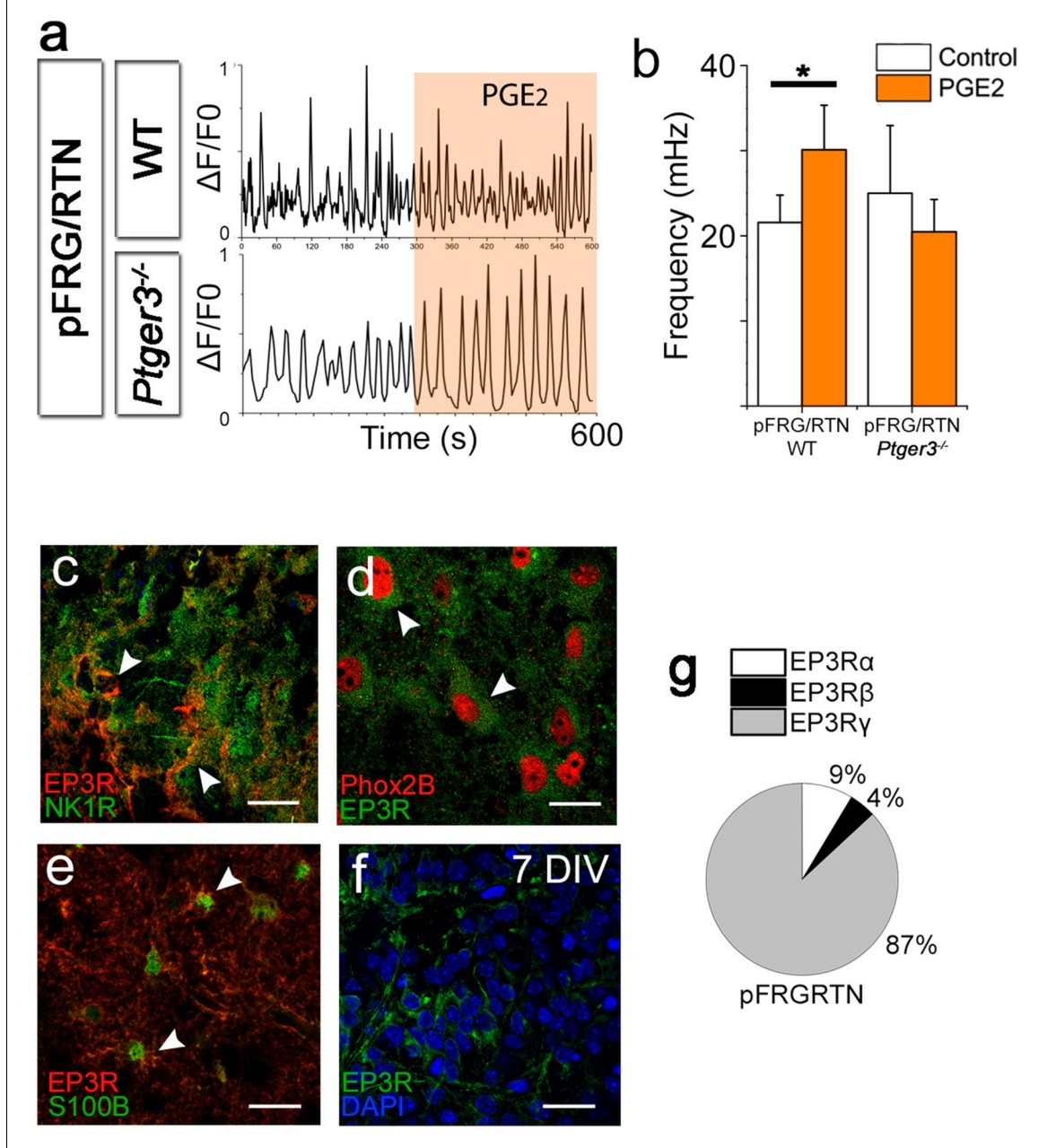

**Figure 12.** PGE$_2$ alters respiratory network activity. In the pFRG/RTN, PGE$_2$ increased the frequency of respiratory (NK1R-expressing) neurons. This PGE$_2$ effect was absent in brainstem slice cultures lacking EP3R (*Ptger3$^{-/-}$*; **a–b**). EP3Rs were present in NK1R-expressing neurons in the pFRG/RTN (**c**, arrowheads, **f**) and co-localized with Phox2b (**d**, arrowheads). EP3Rs were also found on S100B-expressing astrocytes (**e**, arrowheads). Staining was performed on acutely fixed tissue (**c–e**) and brainstem slice cultures (**f**). qRT-PCR showed an abundance of the EP3Rγ (G$_s$-protein coupled) in the pFRG/RTN (N=7; **f**). N: number of slices. DIV: days in vitro. Scale bars: 100 μm. Data are presented as means ± SD. *p<0.05 Source data are available in a separate source data file.

The following source data is available for figure 12:

**Source data 1.** PGE$_2$ frequency data pFRG/RTN.

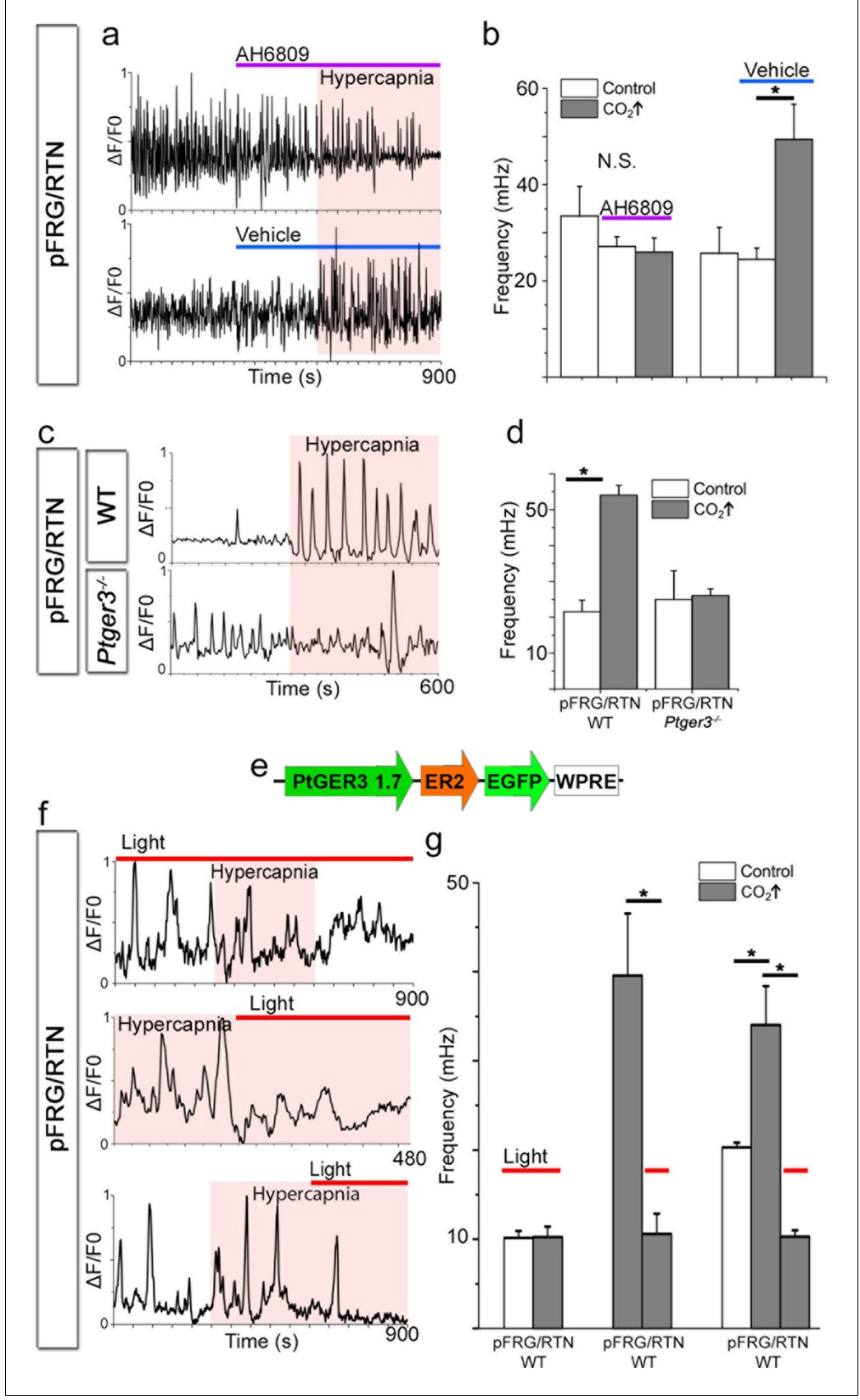

**Figure 13.** PGE$_2$, acting through EP3R, is crucial for the hypercapnic response. Pharmacological inhibition of EP3R by the EP receptor antagonist AH6809 inhibited the response to hypercapnia (increased pCO$_2$[CO$_2$↑]) in the
*Figure 13 continued on next page*

*Figure 13 continued*

pFRG/RTN (N=6, n=472, N.S.; **a–b**). The hypercapnic response was also absent in pFRG/RTN slices lacking EP3R (*Ptger3*$^{-/-}$; N=5, n=348, N.S.; **c–d**). Layout of the lentivirus containing Halo57 (ER2) and eGFP genes under the control of the EP3R promoter (*Ptger3*) used for optogenetics (WPRE=gene enhancing element; **e**). During optogenetic silencing of *Ptger3*-expressing cells, no frequency changes were observed in response to hypercapnia (**f**, top trace; **g**, left graph). The hypercapnic response was also reversed by activating *Ptger3*-Halo57 (**f**, middle and bottom trace; **g**, middle and right graph). Red line: Halo57 activation in response to 625 nm light. N: slices, n: cells. Data are presented as means ± SD. *p<0.05. Source data are available in a separate source data file.

The following source data and figure supplement are available for figure 13:

**Source data 1.** EP antagonist data.
**Source data 2.** Hypercapnia EP3R data.
**Source data 3.** Optogenetics data.
**Figure supplement 1.** Optogenetic silencing of *Ptger3*-expressing cells decreases respiration-related activity.

To further characterize the PGE$_2$ signaling during hypercapnia, we blocked its main receptor, EP3R. Notably, pharmacological blocking of EP receptors (using AH6809, 10 µM) abolished the hypercapnic response (*Figure 13a–b*, 7 DIV), in line with our in vivo data from *Ptger3*$^{-/-}$ mice.

pFRG/RTN slices (7 DIV) from *Ptger3*$^{-/-}$ mice did not respond to hypercapnia (*Figure 13c–d*). Thus, EP3R is important for pFRG/RTN CO$_2$ responsiveness. We next generated a lentiviral vector in which the mouse EP3R (*Ptger3*) promoter controls the expression of the red light-activated halorhodopsin Halo57 fused to eGFP (*Figure 13e*). After transduction, we detected eGFP expression in 90 ± 6% of Phox2b-positive neurons in the pFRG/RTN (*Figure 13—figure supplement 1*). Stimulation by red (625 nm) light of the transduced brainstem slice cultures (7 DIV) triggered hyperpolarization of *Ptger3*-halo57-expressing cells and immediately reduced the calcium signaling frequency of both the network and individual NK1R$^+$ neurons (*Figure 13—figure supplement 1*). This finding indicates a fundamental role for *Ptger3*-expressing cells in the network. Additionally, the response to hypercapnia in the pFRG/RTN was abolished during the light-induced silencing of *Ptger3*-expressing cells. The CO$_2$ response was also reversed by the light-induced halo57 hyperpolarization of *Ptger*-expressing cells (*Figure 13f–g*, *Table 7*).

Based on these findings, we suggest that the PGE$_2$-EP3R pathway is an important mechanism in the hypercapnic response and a modulator of respiratory activity.

## Discussion

Here, we present two novel breathing brainstem organotypic cultures in which the respiration-related preBötC and pFRG/RTN regions maintain their functional organization, activity, and responsiveness to environmental cues. Using these cultures, we show that PGE$_2$ is involved in the control of

**Table 7.** Silencing of *Ptger3*-expressing cells inhibits the response to hypercapnia. Mean frequencies of the pFRG/RTN network during the control period and during exposure to hypercapnia with and without Halo57 stimulation are shown. N.S.: not significant. N: number of slices. Data are presented as mean ± SD.

| N=41 | Mean frequency (mHz) | | |
|---|---|---|---|
| | Control | Hypercapnia | |
| Control | 22.9 ± 9.0* | 34.0 ± 4.3* | p<0.05 |
| Halo57 stimulation | 9.0 ± 1.7 | 10.3 ± 1.1 | N.S. |

sigh activity and the response to hypercapnia via EP3R in the preBötC and the pFRG/RTN, respectively. These findings provide novel insights into central respiratory central pattern generation, its modulation, and the mechanisms underlying breathing disorders during the neonatal period.

Due to the complexity of the respiratory mechanisms, it is difficult to create optimal in vitro model systems that represent in vivo conditions while allowing sufficient depth in detailed mechanisms and their manipulation. The majority of previous studies were performed on brainstem-spinal cord preparations (*en bloc*) (*Onimaru, 1995*) or acute slices (*Ruangkittisakul et al., 2006*). However, these preparations remain active only for hours, making it difficult to study development and long-term effects on respiratory rhythm. Organotypic slice cultures provide a bridge between cell cultures and animals in vivo (*Yamada and Cukierman, 2007*). Their preserved three dimensional structure allows functional circuits to be studied and manipulated over time under microenvironmental control (*Gähwiler et al., 1997*; *Gogolla et al., 2006*; *Yamada and Cukierman, 2007*; *Preynat-Seauve et al., 2009*). First used with hippocampal tissue (*Gähwiler, 1988*), the organotypic culturing method has since expanded to research on the cerebellum (*Lu et al., 2011*) as well as on the brainstem auditory circuits (*Thonabulsombat et al., 2007*). Recently, Phillips and colleagues (*Phillips et al., 2016*) presented an organotypic model system of the preBötzinger complex with respiration-related neuronal rhythm that persists for a month. Here, we characterize this new type of brainstem slice culture further, and also provide details on respiratory network structure and functional respiratory-related motor output. In addition we show that also the pFRG/RTN retains respiration-related rhythmic activity and chemosensitivity. As with all model systems, it has its limitations, e. g., the slices lose several respiratory-related regions (*Smith et al., 2009*). Nonetheless, in contrast to acute slices and the brainstem-spinal cord preparation, our new experimental model system allows long-term studies and manipulation of respiratory networks. This enables the use of different techniques and methods, and significantly reduces the number of procedures that otherwise need to be performed on live animals, as well as the total number of experimental animals. We have exploited this advantage by transfecting the brainstem slice cultures in vitro to be suitable for optogenetic techniques.

Using a newly developed cross-correlation analysis algorithm (*Smedler et al., 2014*), we revealed in the brainstem slice culture, a clustering of cells within the two central pattern generators, a small-world network. A small-world network is characterized by a mean clustering coefficient exceeding that in random networks, but has a mean shortest path-length as short as that in random networks (*Watts and Strogatz, 1998*; *Malmersjo et al., 2013*). Furthermore, the presence of the connective nodes and hubs gives the network a scale-free organization. This finding is in line with a previous topological analysis based on neuronal staining in the preBötC (*Hartelt et al., 2008*). The present insights into the network structure of the pFRG/RTN have not been achieved previously with other methods. Notably, scale-free and small-world networks have been suggested to have evolutionary advantages (*Barabasi and Oltvai, 2004*; *Malmersjo et al., 2013*).

Subsequently we examined how the networks and individual cells were connected. Early in development, gap junctions connect the respiration-related fetal neural networks (*Thoby-Brisson et al., 2009*). During development, gap junction-mediated $Ca^{2+}$-transients stimulate the proliferation of neural progenitor cells (*Malmersjo et al., 2013*) and form a template for chemical synapses to coordinate more mature neural networks (*Jaderstad et al., 2010*). Using CBX and 18-α-GA, we demonstrated that intercellular connections still play a role in postnatal preBötC network activity. This is in line with previous findings (*Elsen et al., 2008*). Notably, even though fewer cells remained active, respiratory neuron frequency and network structure were not affected. Although both CBX and 18-α-GA are commonly used as gap junction inhibitors (*Solomon et al., 2003*; *Elsen et al., 2008*; *Véliz et al., 2008*; *Jaderstad et al., 2010*), these drugs have side effects (*Rekling et al., 2000*; *Schnell et al., 2012*). We used GZA as a control substance because it is structurally similar to CBX but does not have any gap junction inhibiting properties (*Solomon et al., 2003*; *Li and Duffin, 2004*; *Elsen et al., 2008*). However, it mimics many of the side effects of CBX, e.g. the initial stimulatory effect seen in the present study. These limitations need to be kept in mind when interpreting our results on gap junction functions, and further studies are needed to confirm them, preferably using more specific methods of connexin blockage, such as RNAi.

However, our findings do suggest the presence of a neuron-specific subnetwork, connected by chemical synapses, that is able to maintain the network structure. Furthermore, another subnetwork, likely a glial one (*Giaume et al., 2010*; *Okada et al., 2012*) driven by the electrical connections that

modulate network output also seems to be present. Thus, neonatal preBötC synchronization is both gap junction-and synaptic signal-dependent (*Feldman and Kam, 2015*), and it probably contains both neuronal and glial subnetworks. The pFRG/RTN, by contrast, requires gap junctions for its establishment in rodents but is not dependent on them postnatally for rhythmic, correlated network activity (*Fortin and Thoby-Brisson, 2009*). The main mechanism that drives activity in the pFRG/RTN is glutamatergic (*Guyenet et al., 2013*). By contrast, pFRG/RTN gap junctions seem here to be involved in the hypercapnic response (*Figure 10* and *11*). It has been suggested that Cx26 is directly modulated by $CO_2$, independent of $H^+$, through the formation of carbamate bridges (*Meigh et al., 2013*). Our data do not distinguish between intracellular pH-dependent and -independent mechanisms. However, since $PGE_2$ can pass through connexins (*Reyes et al., 2014*), the present data are in line with a $CO_2$-induced, connexin-mediated, release of $PGE_2$ (*Figure 14*).

Prostaglandins are important regulators of autonomic functions in mammals. In many disease states, acute inflammatory responses are initially protective but become harmful under chronic conditions. In our previous reports, we demonstrated how the pro-inflammatory cytokine interleukin (IL)-1β impairs respiration during infection by inducing a $PGE_2$ release in the vicinity of respiratory centers. We also showed that infection is the main cause of respiratory disorders in preterm infants (*Hofstetter et al., 2007*, *2008*) and, in the case of apneas, bradycardias and desaturations (ABD) events in neonates (*Siljehav et al., 2015*). $PGE_2$ is also a key component in the regulation of sigh frequency (*Ramirez, 2014*; *Koch et al., 2015*). During and immediately after birth, $PGE_2$ levels are increased (*Mitchell et al., 1978*). Indeed, the first breaths of extrauterine life are deep and sigh-like,

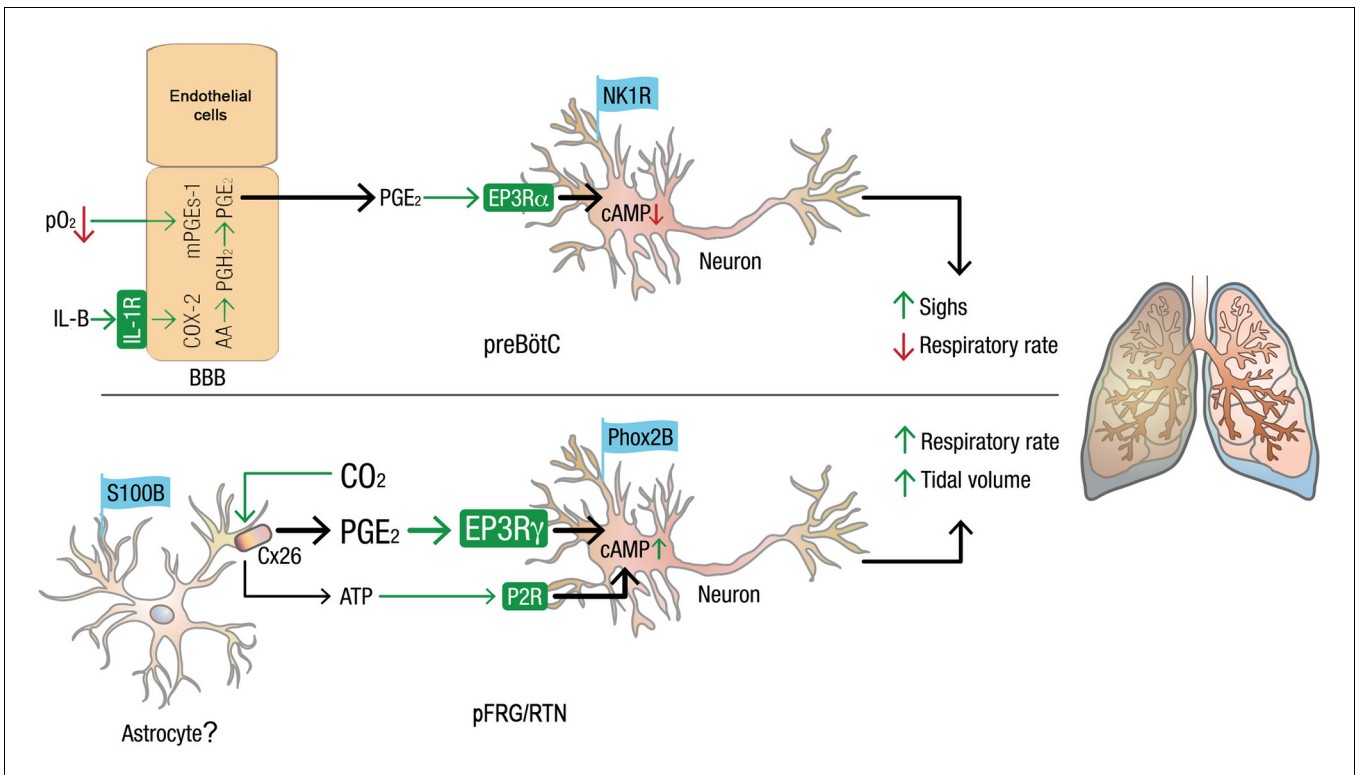

**Figure 14.** Model of how $PGE_2$ modulates respiration and sighs in the preBötC and pFRG/RTN. Systemic inflammation, through the proinflammatory cytokine IL-1β and hypoxia, induces the production of $PGE_2$ in blood brain barrier (BBB) endothelial cells (*Hofstetter et al., 2007*).$PGE_2$ subsequently induces respiratory depression and increases sigh activity via the inhibitory G-protein coupled receptor EP3Rα in the preBötC. In the pFRG/RTN, $PGE_2$ plays a role in the response to elevated $pCO_2$. $CO_2$ directly modulates connexin 26 (Cx26) hemichannels, leading to ATP release. The results in this study suggest that Cx26 also releases $PGE_2$, possiblyfrom mPGEs-1$^+$ astrocytes. $PGE_2$ increases respiratory activity via the stimulatory G-protein coupled receptor EP3Rγ on pFRG/RTN neurons. Thus, inflammation, hypoxia, and hypercapnia alter respiratory neural network and motor output and breathing activity through distinct effects of $PGE_2$ in the pFRG/RTN and the preBötC, respectively. Chronically elevated $PGE_2$ levels, as observed during ongoing inflammation, may decrease the central pattern generators' ability to respond to hypoxic and hypercapnic events. In extreme cases, this decrease may have fatal consequences.

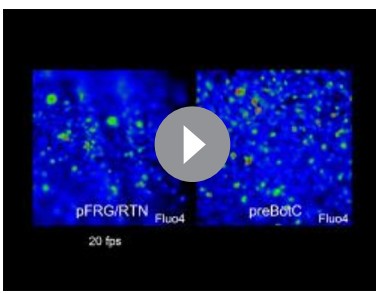

**Video 3.** Parallel display of $Ca^{2+}$ oscillations visualized with Fluo-4 in the pFRG/RTN (left) and preBötC (right). After 15 s, PGE$_2$ (10 nM) is added. This increases the activity of the pFRG/RTN network while the preBötC activity is inhibited. fps: frames per second. For high-resolution versions of the videos, please follow this link to the Karolinska Institutet Cloud Storage system (Box): https://ki.box.com/s/abzuei0yzl4dzbn99995382va6btsq4l.

facilitating alveolar recruitment and $CO_2$ removal (*Mian et al., 2015*). In the brainstem slice cultures, PGE$_2$ had a direct EP3R-dependent effect on both respiratory centers. Notably, PGE$_2$ increased pFRG/RTN but inhibited preBötC frequency (*Video 3*). This finding might be explained by the different distributions of EP3R subtypes in the different regions (*Figure 12*). The coupling to inhibitory or stimulatory G proteins depends on the alternative post-transcriptional splicing of the C-terminal tail of the EP3R preprotein (*Namba et al., 1993*). Furthermore, PGE$_2$ caused a longer $Ca^{2+}$ transient and a higher relative amplitude in an I$_{NaP}$-dependent manner, mimicking the PGE$_2$-based induction of sighs that we observe in vivo and that were recently reported by Koch and colleagues in acute preBötC slices (*Koch et al., 2015*).

Recent data reveal a role of neuromedin B (NMB) and gastrin-related peptide (Grp) and NMB-GPR-expressing preBötC neurons in sighing (*Li et al., 2016*). In addition to these peptidergic pathways, the present and recent data from Koch and colleagues (*Koch et al., 2015*) suggest that low concentrations of the inflammation-associated PGE$_2$ induce sighs, acting through modulation of the persistent sodium current in preBötC neurons.

The preBötC results presented in this study provide evidence for how the general respiratory depression induced by inflammatory signaling, previously reported in vivo and in vitro (*Hofstetter et al., 2007*) and in human neonates (*Hofstetter et al., 2007*; *Siljehav et al., 2015*), is mediated by a direct effect of PGE$_2$ on EP3R (*Siljehav et al., 2012*) in the preBötC. The present data may help to further explain the mechanism underlying apneas that occur during infectious periods in neonates (*Hofstetter et al., 2007*, *2008*; *Di Fiore et al., 2013*; *Siljehav et al., 2015*).

Another common respiratory problem in neonates, particularly premature infants, is an inability to respond adequately to hypoxia and hypercapnia. This may cause recurrent hypoxia, leading to cognitive disabilities later in life (*Greene et al., 2014*). A disruption of central $CO_2$ chemosensitivity is commonly seen in children with bronchopulmonary dysplasia (*Di Fiore et al., 2013*), leading to chronic hypoventilation, which may explain why these infants have an increased risk of sudden infant death syndrome (*Martin et al., 2011*). Therefore, we investigated the role of the pFRG/RTN in chemosensitivity (*Guyenet et al., 2013*) and found that the response to hypercapnia is dependent on functioning gap junctions. This is in line with previous findings showing that Cx26 is directly modified by $CO_2$ (*Meigh et al., 2013*).

These $CO_2$-sensitive connexin hemichannels can release ATP, and indeed the hypercapnic response is partly mediated by purinergic type 2 receptors (*Erlichman et al., 2010*; *Gourine et al., 2010*; *Guyenet et al., 2013*). In addition to these purinergic pathways, we suggest that EP3R-dependent signaling is involved in the response to altered pCO$_2$. Genetic ablation of *Ptger3* reduced the hypercapnic response both in vivo and in vitro, as did pharmacological blockage in vitro, in line with our previous experiments (*Siljehav et al., 2014*). Moreover, the optogenetic inhibition of *Ptger3*-expressing cells in the pFRG/RTN revealed that these cells are essential for the $CO_2$ response. We also demonstrated that PGE$_2$ is released during hypercapnic exposure, likely through Cx26 or other $CO_2$-sensitive connexins (*Huckstepp et al., 2010b*). Thus, part of the $CO_2$ response seems to be mediated by a gap junction-dependent release of PGE$_2$.

Generation of active expiration is another important function of the pFRG/RTN (*Feldman et al., 2013*). It is possible that PGE$_2$ stimulates both chemosensitive neurons and neurons important for active expiration. Such neuronal populations could overlap, but the ventral part pFRG/RTN seems to have a more chemosensitive character while the lateral part displays rhythmic activity and enforces active expiration when stimulated (*Pagliardini et al., 2011*; *Feldman et al., 2013*; *Huckstepp et al., 2015*). The $CO_2$-sensing of the pFRG/RTN slice remains functional. Whether the rhythmic activity we

observe in the pFRG/RTN is generated by "active expiration-neurons" is outside the scope of the present study. Future studies should aim to investigate whether $PGE_2$ also may affect active expiration.

The pFRG/RTN is the best-recognized central chemosensitive region. However, in our pFRG/RTN brainstem slice culture, neurons of the raphe nucleus should be present (*Smith et al., 2009*). Such neurons may also have chemosensing properties (*Richerson, 2004*), though this has not been shown conclusively (*Depuy et al., 2011*). From the raphe nucleus there are evidence of projections to the pFRG/RTN (*Guyenet et al., 2009*), and we cannot exclude the possibility that these are preserved in the brainstem slice culture.

The effects of $CO_2$ in the present study are based on a change in carbamylation of specific proteins, e.g. Cx26 (*Meigh et al., 2013*), or intracellular pH, but testing these alternatives goes beyond the scope of the present work. In our experimental setup the extracellular pH remained stable while the dissolved $CO_2$ increased. This specific approach was selected because $CO_2$ has a direct modulating effect on connexins, allowing passage of small molecules (*Huckstepp et al., 2010a*; *Huckstepp and Dale, 2011*; *Meigh et al., 2013*), and our hypothesis was that $PGE_2$ is released through such connexins.

What still remains to be determined the exact source of the $PGE_2$ released during hypercapnia. The indication of a gap junction-dependent release of $PGE_2$ together with the presence of mPGEs-1 in pFRG/RTN astrocytes suggests that the $PGE_2$ is of astrocytic origin. This would be in line with previous findings of astrocytic ATP release during hypercapnia (*Gourine et al., 2010*; *Huckstepp et al., 2010a*). The astrocytic involvement in the $CO_2$ response is also evident in a Rett syndrome model (methyl-CpG-binding protein 2 (MeCP2) knockout), in which conditional MeCP2 knockout in astroglia blunts the $CO_2$ response (*Turovsky et al., 2015*). We think that mPGEs-1-expressing astrocytes are the likely source, even though alternative sources of $PGE_2$, such as endothelial cells or microglia, remain to be investigated with regards to their possible involvement in the pFRG $CO_2$ response. Nonetheless, $CO_2$-mediated $PGE_2$ release introduces a novel chemosensitive pathway (*Figure 14*).

As $PGE_2$ and the EP3R are directly involved in and modulate both the respiratory rhythm-generating preBötC and the Phox2b chemosensitive neurons, $PGE_2$ from other sources, such as endothelial cells during hypoxia and inflammation (*Hofstetter et al., 2007*), will alter the hypercapnic and the hypoxic responses. $PGE_2$ has prominent respiratory depressant effects in humans, sheep, pigs, and rodents (*Guerra et al., 1988*; *Long, 1988*; *Ballanyi et al., 1997*; *Hofstetter et al., 2007*; *Siljehav et al., 2015*). The $PGE_2$-induced attenuation of these vital brainstem neural networks, e.g., during an infectious response, could result in gasping, autoresuscitation failure and ultimately death. However, how chronic $PGE_2$ release associated with ongoing inflammation alters plasticity and the responsiveness to $CO_2$ must be further investigated.

To conclude, we identified a novel pathway in the hypercapnic response of brainstem neural networks that control breathing. This pathway depends on EP3R and gap junctions and is partly mediated by the release of $PGE_2$, linking chemosensitivity control to the inflammatory system. The present findings have important implications for understanding why and how ventilatory responses to hypoxia and hypercapnia are impaired and inhibitory reflexes exaggerated in neonates, particularly during infectious episodes.

## Materials and methods

### Subjects

C57 black (C57BL/6J) inbred mice (Charles River, Wilmington, MA) were utilized in the experiments. The eicosanoid prostanoid 3 receptor (EP3R) gene (*Ptger3)* was selectively deleted in knockout mice ($Ptger3^{-/-}$) with a C57BL/6J background, as described preciously (*Fabre et al., 2001*). C57BL/6J mice were then used as experimental controls for $Ptger3^{-/-}$ mice. As results from $Ptger3^{-/-}$ mice were consistent with pharmacological and optogenetic inhibition of EP3Rs, we can confirm the lost EP3R function in the mice.

To determine the location of mPGEs-1, mice expressing green fluorescent protein (GFP) under the GFAP promoter were used. Frozen sperm from the GFAP-tTA (*Lin et al., 2004*; *Pascual et al., 2005*) and tetO-Mrgpra1 (*Fiacco et al., 2007*) mouse strains were purchased from the Mutant

Mouse Regional Resource Centers supported by NIH (MMRRC). The strains were re-derived by Karolinska Center for Transgene Technologies (KCTT), and the offspring was crossed as previously described (*Fiacco et al., 2007*). Double transgenics were identified by PCR according to MMRRC's instructions.

All mice were reared by their mothers under standardized conditions with a 12:12-hr light-dark cycle. Food and water was provided *ad libitum*. The studies were performed in accordance with European Community Guidelines and approved by the regional ethic committee. The animals were reared and kept at the Department of Comparative Medicine, Karolinska Institutet, Stockholm, Sweden.

## Dual-chamber plethysmography in vivo

Ventilatory measurements were made using dual-chamber plethysmography in 9-day old (P9) mice. Mice were cooled on ice for 2–3 min and then prostaglandin E$_2$ (PGE$_2$, 1 µM; Sigma-Aldrich, St. Louis, MO, USA, cat no. P5640) or vehicle (artificial cerebrospinal fluid, aCSF, containing in mM: 150.1 Na$^+$, 3 K$^+$, 2 Ca$^{2+}$, 2 Mg$^{2+}$, 135 Cl$^-$, 1.1 H$_2$PO$_4^-$, 25 HCO$_3^-$ and 10 glucose) was slowly injected into the lateral ventricle by using a thin pulled glass pipette attached to polyethylene tubing (*Siljehav et al., 2014*). The mouse was then immediately placed into the plethysmograph chamber. After a 10-min recovery period, confirming stable respiration and body temperature, respiratory parameters in normocapnia (air) was established followed by a hypercapnic challenge (5% CO$_2$ and 20% O$_2$ in N$_2$) for 5 min. This was followed by 5 min of normocapnia. Skin temperature was measured throughout experimentation and remained stable. After experimentation, the mice were anesthetized with 100% CO$_2$ and decapitated. The brain was dissected and examined at the injection site and for the presence of any intracranial hemorrhage. Three of 28 animals had visible intracranial bleeding and were excluded from analysis.

## Brainstem organotypic culture

P3 mice pups were used for the establishment of brainstem organotypic slice cultures. The pups were decapitated at the cervical C3–C4 level. The heads were washed with cold dissection medium consisting of 55% Dulbecco's modified Eagle's medium (Invitrogen, Paisley, UK), 0.3% glucose (Sigma-Aldrich, St. Louis, MO, USA), 1% HEPES buffer (Invitrogen, UK) and 1% Antibiotic-Antimycotic (Invitrogen, UK). After washing, the heads were moved to fresh dissection medium on ice. The entire brain was dissected. During dissection, extra caution was taken around the cerebellopontine angle to ensure that the respiratory regions of the brainstem were not damaged. Nerves were cut with microscissors.

The brain was sectioned into 300-µm-thick transverse slices by using a McIlwain Tissue Chopper (Ted Pella, Inc., Redding, CA, USA). Slices were selected by using anatomical landmarks, such as the shape and size of the entire slice and the fourth ventricle. For location of the preBötzinger complex (preBötC), the presence of nucleus hypoglossus, nucleus spinalis nervi trigemini, pyramis medullae oblongatae and nucleus tractus solitarius (not always clearly seen), together with the absence of the anterior horn for the nucleus cochlearis, according to online references (*Ruangkittisakul et al., 2006*, *2011*, *2014*). For location of the parafacial respiratory group/retrotrapezoid nucleus (pFRG/RTN), the presence of the nucleus facialis was used. On the slices, the preBötC is located within ventrolateral regions, and the pFRG/RTN is located at the ventrolateral edge.

Selected slices were washed by moving them to brain slice medium (55% Dulbecco's modified Eagle's medium, 32.5% Hank's balanced salt solution, 0.3% glucose, 10% fetal bovine serum, 1% HEPES buffer and 1% Antibiotic-Antimycotic [Invitrogen, UK]), after which they were carefully placed on insert membranes (Millicell Culture Plate Inserts; Millipore, Billerica, MA, USA) in six-well plates. The membranes were coated in advance with poly-L-lysine (0.3 ml; 0.1 mg/ml, Sigma-Aldrich, St. Louis, MO, USA). Brain slice medium (1 ml) was placed underneath the membrane, and all fluid on top of the membrane was removed. It is important not to cover the slices with medium, because this may impair oxygenation (*Frantseva et al., 1999*). The brainstem slice cultures were maintained in an incubator (37°C, 5% CO$_2$), and the brain slice medium was changed every second day. The brainstem slices were kept in culture for 7–21 days in vitro (DIV) before fixation or live imaging experiments. For a detailed protocol, see Herlenius and colleagues (*Herlenius et al., 2012*).

## Immunohistochemistry

For immunohistochemistry, brainstem slice cultures were fixed with cold paraformaldehyde (4%) in PBS for 1 hr at 4°C and 20% ice-cold methanol in PBS for 10 min. Permeabilization was conducted by using 0.2% Triton X-100 (Roche Diagnostics, Hofgeismar, Germany) and 0.1% Tween 20 (Invitrogen, UK) in PBS for 40 min at room temperature (RT). Thereafter, slices were blocked in 5% bovine serum albumin (BSA; Invitrogen, UK) and 0.05% Tween 20 in PBS for 2 hr at RT. The Millicell insert membranes were carefully cut with a scalpel and placed back into the wells. The primary antibodies were diluted 1:200 in 0.05% Tween 20/PBS and incubated at 4°C for 48 hr. Next, the slices were washed 3 × 10 min with PBS and incubated for 1.5 hr at RT with Alexa Fluor-conjugated secondary antibodies (Invitrogen, UK) diluted 1:200 in 0.05% Tween 20/PBS. The slices were then washed 3 × 10 min with PBS and mounted in ProLong Gold Antifade Reagent with DAPI (Invitrogen, UK, cat. no. P36931). Primary antibodies used were mouse anti-microtubule associated protein 2 (MAP2; Invitrogen, cat. no. P11137), rabbit anti-neurokinin 1 receptor (NK1R; Sigma-Aldrich, St. Louis, MO, USA, cat no. S8305), mouse anti-GFAP (Chemicon, Temecula, CA, USA, cat no. MAB360), rabbit anti-S100β (Millipore; cat. no. 04–1054), mouse anti-neuron-specific class III β-tubulin (Tuj1; Covance, Princeton, NJ, USA, cat no. MMS-435P), rabbit anti-K$^+$/Cl$^-$ cotransporter 2 (KCC2; Millipore, cat no. 07–432), rabbit anti-vesicular glutamate transporter 2 (VGLUT2; Synaptic Systems, Goettingen, Germany, cat no. 135–402), mouse anti-connexin 26 (Cx26; Invitrogen, Inc., San Francisco, CA, cat no. 13–8100), rabbit anti-connexin 32 (Cx32; Invitrogen, cat. no. 71–0600), mouse anti-connexin 43 (Cx43; Zymed, cat no 13–8300), goat anti-Phox2b (Santa Cruz Biotechnology, Santa Cruz, CA, USA, cat no 13224), goat Phox2b antibody (R & D Systems, Minneapolis, MN, USA), and rabbit anti-caspase 3 (Cell Signaling Technology, Beverly, MA, USA, cat no. 9661). Negative controls with only secondary antibodies showed no staining.

For EP3R staining, a different protocol was used. Initially, brains were fixed with 4% paraformaldehyde overnight followed by 10% sucrose overnight and then frozen to -80%. The frozen brainstems were cryosectioned and blocked in blocking buffer (1% BSA, 5% donkey serum, 5% dimethyl sulfoxide (DMSO), 1% Triton X-100 in Tris-buffered saline (TBS, consisting of 6 mM Tris-HCl, 1 mM Tris base and 9 mM NaCl in ddH$_2$O) for 1 hr at RT. After blocking, the slices were incubated with polyclonal rabbit anti-EP3R antibody (Cayman Chemical Co., Ann Arbor, MI, USA) diluted 1:50 in 10% DMSO containing 0.2% Triton X-100 in TBS at RT overnight. Next, slices were washed 3 × 15 min with TBS with agitation, followed by incubation for 1 hr in the dark with Alexa Fluor 488-conjugated donkey anti-rabbit secondary antibody (Life Technologies, Grand Island, NY, USA) diluted 1:1000 in 1% BSA, 2% donkey serum, 2% DMSO and 5% Triton X-100 in TBS. The slices were then washed 3 × 15 min with TBS with agitation, and blocked again for 1 hr at RT in the same blocking buffer as used previously. After blocking, the slices were incubated with the second primary antibody, diluted 1:200 in 10% DMSO containing 0.2% Triton X-100 in TBS at 4°C overnight. Following overnight incubation, the slices were washed 3 × 15 min with TBS with agitation and incubated with Alexa Flour 647-conjugated donkey anti-goat secondary antibody (Life Technologies, Grand Island, NY, USA) diluted 1:1000 in 1% BSA, 2% donkey serum, 2% DMSO and 5% Triton X-100 in TBS. Finally, the slices were washed 3 × 15 min with TBS with agitation, and mounted in ProLong Gold Antifade Reagent with DAPI.

Antibody binding was controlled by including an irrelevant rabbit polyclonal IgG isotype control (Bioss, Woburn, MA, USA). EP3R staining was controlled by including an EP3R blocking peptide reconstituted in distilled water mixed with EP3R antibody at a 1:1 (v/v) ratio. A pre-incubation of EP3R antibody with the blocking peptide for 1 hr at RT was necessary before the antibody was added to the slice. The peptide was used in conjunction with the antibody to block protein-antibody complex formation during immunohistochemical analysis for the EP3Rs. These controls showed no staining.

Double immunofluorescence staining was also performed according to Westman and colleagues (*Westman et al., 2004*) using polyclonal rabbit anti-human microsomal prostaglandin E synthase 1 antiserum (mPGES-1; Cayman chemicals, cat. no. 160140) and monoclonal anti-mouse glial fibrillary acidic protein antibody (GFAP; Chemicon, Temecula, CA, USA, cat no. MAB360). PBS supplemented with 0.1% saponin (PBS-saponin) was used as a buffer through the experiment. Endogenous peroxidase activity was blocked using PBS containing 1% H$_2$O$_2$ and 0.1% saponin for 60 min in darkness. Endogenous biotin was blocked using an avidin-biotin blocking kit (Vector

Laboratories, Burlingame, CA) supplemented with 0.1% saponin. The sections were incubated with primary antibodies overnight, in PBS-saponin containing 3% BSA antibody solution. Thereafter, they were blocked with 1% normal goat serum, or normal donkey serum (depending on the host of secondary antibody) in PBS-saponin for 15 min, followed by 1-hr incubation with secondary antibody, donkey anti-rabbit alexa fluorophore 488 or goat anti-mouse Alexa Fluor 546.

## Propidium iodide staining

Propidium iodide (1 ml/L, Invitrogen, UK) was added to brain slice medium (dilution 1:1000). Staining solution (1 ml) was added on top of the membrane with the brainstem slice cultures and incubated at 37°C (5% $CO_2$) for 3 hr. Immediately after incubation, the brainstem slice cultures were fixed in 4% paraformaldehyde for 1 hr. Positive controls were made by first treating the brainstem slice culture for oxygen glucose deprivation (OGD) for 1 h, as described by Montero Dominguez and colleagues (*Montero Domínguez et al., 2009*).

## Electrophysiology

Whole-cell patch recordings were obtained from brainstem slice cultures at a temperature of 34°C. Cells were visualized by using IR-differential contrast microscopy (Axioskop FS, Carl Zeiss, Jena, Germany). Recorded cells were selected visually, and paired recordings were obtained for neurons with lateral somatic distances of <100 µm. Recordings were amplified by using 700B amplifiers (Molecular Devices, Sunnyvale, CA, USA), filtered at 2 kHz, digitized at 5–20 kHz by using ITC-18 (Instrutech, Longmont, CO, USA), and acquired by using Igor Pro (Wavemetrics, Lake Oswego, OR, USA). Patch pipettes were pulled with a P-97 Flamming/Brown micropipette puller (Sutter Instruments, Novato, CA, USA) and had an initial resistance of 5–10 MΩ in a solution containing in mM: 110 K-gluconate, 10 KCl, 10 HEPES, 4 Mg-ATP, 0.3 GTP and 10 phosphocreatine. Recordings were performed in current-clamp mode, with access resistance compensated throughout the experiments. Recordings were discarded when access resistance increased beyond 35 MΩ. To characterize the electrical properties of the recorded cells, depolarizing and hyperpolarizing current steps and ramps were injected, enabling the extraction of properties such as input resistance, membrane time constant and action potential threshold. Electrophysiological properties were presented as box plots, with maximum and minimum values.

For recording of hypoglossal nerve activity and hypoglossal nucleus neuronal population discharge, an extracellular suction electrode was used together with a Model 1700 AC amplifier (A-M systems, Carlsborg, WA, USA) and AxoScope software, version 9.2 (Axon Instruments, Union City, CA, USA). Recordings were made with a sampling interval of 0.3 ms.

## $Ca^{2+}$ time-lapse imaging

For $Ca^{2+}$ imaging, Fluo-4 AM (Invitrogen, UK) dissolved in DMSO (Invitrogen, UK) was used at 10 µM in serum free brain slice medium or artificial cerebrospinal fluid (aCSF, containing in mM: 150.1 $Na^+$, 3 $K^+$, 2 $Ca^{2+}$, 2 $Mg^{2+}$, 135 $Cl^-$, 1.1 $H_2PO_4^-$, 25 $HCO_3^-$ and 10 glucose) together with 0.02% pluronic acid (Invitrogen, UK). We did not observe any differences in $Ca^{2+}$ activity between HEPES-free brain slice medium and aCSF during $Ca^{2+}$ imaging, despite slight differences in $[K^+]$ and $[Ca^{2+}]$, which both affect the rhythm of the slice (*Ballanyi and Ruangkittisakul, 2009*). A higher $[Ca^{2+}]$ or an increase in $[K^+]$ from 3 mM to 4.8 mM did not affect the network properties in our system. To localize the preBötC or the pFRG/RTN, tetramethylrhodamine-conjugated Substance P (TMR-SP; Biomol, Oakdale, NY, USA) was used at a final concentration of 3 µM in brain slice medium or aCSF. The TMR-SP solution was placed on top on the brainstem slice and incubated for 10–12 min at 37°C in an atmosphere of 5% $CO_2$. The TMR-SP solution was then replaced with 1 ml of 10 µM Fluo-4 solution. The Fluo-4 solution was incubated for 30–40 min (37°C, 5% $CO_2$). Before imaging, the slice was washed with brain slice medium/aCSF for 10 min (37°C, 5% $CO_2$).

During time-lapse imaging, slices were kept in an open chamber perfused with HEPES-free brain slice medium (containing in mM: 132 $Na^+$, 4.8 $K^+$, 1.4 $Ca^{2+}$, 0.74 $Mg^{2+}$, 112 $Cl^-$, 0.76 $H_2PO_4^-$, 25.6 $HCO_3^-$ and 16.8 glucose) or aCSF (2.5 ml/min) by using a peristaltic pump. A Chamlide Inline Heater (Live Cell instruments, Seoul, Korea, cat no. IL-H-10) was used for temperature control, and a Chamlide AC-PU perfusion chamber for 25-mm coverslips (Live Cell instruments, Seoul, Korea, cat no. ACPU25) was used for perfusion. HEPES-free medium was used to minimize the risk for hydrogen

peroxide formation (*Lepe-Zuniga and Gery, 1987*). The medium or aCSF was constantly bubbled with 5% $CO_2$ and 95% $O_2$. The temperature of the chamber was set to 32°C, which Hartelt and colleagues (*Hartelt et al., 2008*) showed to be well tolerated by neurons. Images were captured by using a Zeiss AxioExaminer D1 microscope equipped with 20× and 40× water immersion objectives (N.A. 1.0), a monochromatic Zeiss MrM CCD-camera, a Photometrics eVolve EMCCD-camera and filter sets 38HE (Zeiss), 43 (Zeiss), and et560/hq605 (Chroma, Bellows Falls, VT, USA). For live imaging, a frame interval of 0.1–2 s was used. Exposure time was set to 100–300 ms.

Substances added during imaging were [D-Ala2, N-Me-Phe4, Gly5-ol]-enkephalin (DAMGO, 0.5 µM; Sigma-Aldrich, St. Louis, MO, USA, cat no. E7384), carbenoxolone (CBX 50, µM; Sigma-Aldrich, St. Louis, MO, USA, cat no. C4790), 18α-glycyrrhetinic acid (18-α-GA, 25 µM; Sigma-Aldrich, St. Louis, MO, USA, cat no. G10105), glycyrrhizic acid (GZA, 50 µM; Sigma-Aldrich, St. Louis, MO, USA, cat no. 50531), tetrodotoxin (TTX, 20 nM, Abcam, Cambridge, UK, cat.no. 120055), riluzole (10 µM, Sigma-Aldrich, St. Louis, MO, USA, cat.no. R116), flufenamic acid (FFA, 50 µM, Sigma-Aldrich, St. Louis, MO, USA, cat.no. F9005), Suramin (100 µM; Sigma-Aldrich, St. Louis, MO, USA, cat no. S2671), TNP-ATP (20 nM; Sigma-Aldrich, St. Louis, MO, USA, cat. no. SML0740), AH6809 (Cayman Chemicals, cat.no. 33458-93-4) and prostaglandin E2 ($PGE_2$, 10 nM; Sigma-Aldrich, St. Louis, MO, USA, cat no. P5640). All substances were dissolved in brain slice medium/aCSF prior to experimentation and added to the chamber by using a continuous flow system. For each experiment, a control period with regular medium/aCSF was followed by drug application. GZA was used as a negative control for the gap junction inhibitors CBX and 18α-GA because it has non-gap junction-inhibiting properties, but similar side effects to those of CBX. Specificity was tested by using a second batch of medium or aCSF. During infections in neonatal children, $PGE_2$ is present at a concentration of 15 pM in cerebrospinal fluid (*Hofstetter et al., 2007*). A higher concentration (10 nM) was used to compensate for the in vivo metabolism of the molecule.

Exposure to isohydric hypercapnia was done by using aCSF adjusted with a high bicarbonate buffer concentration (in mM: 150.1 $Na^+$, 3 $K^+$, 2 $Ca^{2+}$, 2 $Mg^{2+}$, 111 $Cl^-$, 1.1 $H_2PO_4^-$, 50 $HCO_3^-$ and 10 glucose). This generated a hypercapnic carbon dioxide partial pressure ($pCO_2$) of 6.6 kPa at pH 7.5 when aCSF was saturated with 8% $CO_2$.

## Viral transfection and optogenetics

A subgroup of 1-DIV-old brainstem slices were moved to a separate BSL-2 laboratory where they were transduced with a mouse prostaglandin E receptor 3 (subtype EP3) lentivirus (*Ptger3*) containing Halo57, developed in collaboration with Dr Robert Finney (Xactagen, Shoreline, WA, USA), by applying 0.2 µl of virus suspension on top of the slice. The brainstem slice cultures were then placed in an incubator for 5 days, and after washing with warm brain slice medium at time points 2 and 5 days, the brainstem slice cultures were moved back to the original laboratory and placed in an incubator overnight. $Ca^{2+}$ time-lapse imaging was performed on the slices as described above. Halo57 was stimulated continuously during $Ca^{2+}$ time-lapse imaging by using a 625-nm LED in a custom-built system (Thorlabs, Newton, NJ, USA).

The optogenetically inhibited network and NK1R positive neurons retained their response to general depolarization induced by elevated $[K^+]$ (Supplementary Fig. S6).

## $PGE_2$ ELISA

The release of $PGE_2$ in aCSF during control and hypercapnic conditions was assessed by ELISA. The aCSF samples were collected through perfusion system, during control and hypercapnic period and either analyzed immediately or stored at -80°C. For the validation of the experiments, two different ELISA kits have been used.

Prostaglandin $E_2$ EIA monoclonal kit by Cayman Chemical (Ann Arbor, MI, US) was performed according to standard procedure. Firstly, the $PGE_2$ EIA Standard was prepared from #1 to #8. The 96-well plate was ready to use and contained a minimum of two blanks (Blk), two non-specific binding wells (NSB), two maximum binding wells (B0) and an eight point standard curve run in duplicate. Each sample was assayed in triplicate. The 96-well plate was coated for 18 hr at 4°C with 50 µl of Prostaglandin $E_2$ AChE Tracer and 50 µl of Prostaglandin $E_2$ Monoclonal Antibody per well. Plate was washed three times with specific Wash Buffer and in consequence, it was developed in the dark

at room temperature on a plate shaker for 60–90 min by adding 200 µl of Ellman's Reagent to each well. Finally, the plate was read at 405 nm.

PGE$_2$ ELISA kit by Enzo Life Sciences (Farmdale, NY, US) was also used for the confirmation of the results. A similar process was followed but a bit shorter. Samples were assayed in duplicate. The 96-well plate was incubated at room temperature on a plate shaker for 2 hr with 50 µl of PGE$_2$ conjugate and 50 µl of antibody solution per well. Then, the plate was washed three times with washing solution. After the wash, 200 µl of the pNpp substrate solution were added to every well and the plate was incubated at room temperature for 45 min. Finally, 50 µl of Stop Solution were added to every well in order to stop the reaction and the plate was read immediately at 405 nm.

## Quantitative real-time PCR

The preBötC and pFRG/RTN regions were cut out from brainstem slices with micro scissors. The samples were pooled together litterwise to minimize the effect of different tissue piece sizes, and provide enough cells for accurate analysis. RNA was isolated from the tissue samples using the miR-CURY RNA isolation Kit (Exiqon) according to manufacturer's instructions. cDNA was synthesized from 20 ng RNA using SuperScript VILO cDNA Synthesis Kit (Invitrogen). The reverse transcription was performed according to the manufacturer's protocol. Real-time PCR was run with Power SYBR Green PCR Master Mix (Applied Biosystems) and amplified in a 7500 Real Time PCR system (Applied Biosystems). Primers are listed in *Table 8*. As endogenous control, glucose-3 phosphate dehydrogenase (GAPDH; Applied Biosystems) was used. Relative quantification (RQ) values were calculated using the CT$^{(\Delta\Delta CT)}$ method (*Livak and Schmittgen, 2001*).

## Data analysis

From in vivo plethysmograph recording (LabChart Pro, v 8.0.10, AD Instruments, Dunedin, New Zealand), periods of calm respiration without movement artifacts were selected for analysis based upon visual observations during experimentation as in previous studies (*Hofstetter and Herlenius, 2005*). Mean respiratory frequency (F$_R$; breaths/min), tidal volume (V$_T$) and minute ventilation (VE) during normocapnic and hypercapnic periods were calculated as described previously (*Hofstetter and Herlenius, 2005*). Sighs were excluded from the analysis. V$_T$ and VE were divided by body weight (BW) and expressed as milliliters per gram and milliliters per gram per min, respectively. The number of sighs, defined as breath with larger amplitude and a biphasic inspiratory phase, was calculated manually and expressed as sighs per min.

Immunohistochemical staining was analyzed in a Zeiss AxioExaminer D1 microscope (10×, 20× and 40× water immersion objectives) or a Zeiss LSM700 confocal (40× and 63× oil-immersion objectives), and captured images were processed by adjusting contrast in ImageJ (1.42q, National Institutes of Health, Bethesda, MD, USA) to reduce background staining.

Ca$^{2+}$ imaging time traces were analyzed with a recently published method (*Malmersjo et al., 2013*; *Smedler et al., 2014*). Regions of interest were marked for all cells based on the standard deviation of fluorescence intensity over time, by using a semiautomatic-adapted ImageJ script kindly provided by Dr. John Hayes (The College of William and Mary, Williamsburg, VA, USA, http://physimage.sourceforge.net/). The mean intensity value and coordinates were measured using ImageJ. Average intensities of regions of interest were quantified for each frame, and dynamic fluorescence signals were normalized to baseline values. The linear similarity (Pearson

**Table 8.** Primers used for qRT-PCR.

| Oligo name | Sequence |
| --- | --- |
| EP3alfa forward<br>EP3alfa reverse | GCTTCCAGCTCCACCTCCTT<br>CATCATCTTTCCAGCTGGTCACT |
| EP3 sense<br>EP3beta anti-sense | 5'-TGACCTTTGCCTGCAACCTG-3'<br>5'-GACCCAGGGAAACAGGTACT-3' |
| EP3gamma forward<br>EP3gamma reverse | AGTTCTGCCAGGTAGCAAACG<br>GCCTGCCCTTTCTGTCCAT |

**Table 9.** Successful experiments behind representative images.

| Figure | Panel | Number of experiments |
|---|---|---|
| 1 | c | 25 |
| 2 | b | 31 |
| | c | 23 |
| | d | 9 |
| | e | 19 |
| | f | 14 |
| | g | 112 |
| 2 – S1 | | 112, 23, 12 |
| | | 33, 23, 10 |
| | | 27, 12, 11 |
| | | 22, 15, 8 |
| | | 15, 8, 8 |
| 2 – S2 | | 5 |
| 2 – S3 | a | 12 |
| | b | 5 |
| | c | 20 |
| 3 | a | 5 |
| | b | 5 |
| 3 – S1 | a | 5 |
| | b | 5 |
| 4 | b | 12 |
| | c | 13 |
| | d | 8 |
| | g | 9 |
| | h | 5 |
| 4-S2 | | 840, 621, 456 |
| 5 | a | 16 |
| 5 – S1 | a | 6 |
| 5 – S2 | a | 11 |
| | c | 5 |
| 6 | a | 9 |
| | b | 8 |
| | c | 8 |
| | d | 3 |
| | e | 8 |
| | f | 6 |
| | g | 7 |
| | h | 8 |
| | i | 6 |
| | j | 7 |
| 7 | a | 5, 4 |
| | c | 9 |
| | d | 12 |
| | f | 5, 4 |
| 7 – S1 | | 4, 4 |
| 8 | b | 9 |
| | c | 9 |
| | d | 9 |
| | e | 5 |
| 8 – S1 | | 35, 22, 19 |
| | | 9, 9 |
| | | 5 |
| 9 | b | 12 |
| | c | 11 |
| | d | 6 |
| 9 – S1 | | 315, 429, 192 |
| 9 – S2 | | 18, 7 |

*Table 9 continued on next page*

*Table 9 continued*

| Figure | Panel | Number of experiments |
|---|---|---|
| 10 | a | 7 |
| | b | 7, 7 and 5 |
| 11 | a | 12, 4 |
| 11 – S1 | a | 11 |
| | b | 11 |
| | c | 6 |
| 12 | a | 5, 4 |
| | c | 16 |
| | d | 44 |
| | e | 6 |
| | f | 12 |
| 13 | a | 6, 5 |
| | c | 7, 5 |
| | f | 41 |
| 13 – S1 | a | 6 |
| | b | 27 |

correlation) was calculated (*Figure 4—figure supplement 1*) between pairs of $Ca^{2+}$ traces with a custom-made script in MATLAB (version 7.9.0.529 R2009b; MathWorks, Natick, MA, USA) and by using the mic2net toolbox (*Smedler et al., 2014*) (version 6.12; MathWorks). Calculating the pairwise correlation coefficients resulted in a correlation matrix that was converted to an adjacency matrix by applying a cut-off level. The cut-off level was selected by calculating the mean of the $99^{th}$ percentile of correlation coefficients for a set of experiments with scrambled signals. Scrambling was performed by randomly translating all traces in the time-domain. The network structure was visualized by plotting a line between pairs of cells, where the color of the lines was proportionate to the correlation coefficient. This was plotted on top of an image of the standard deviations of the fluorescence over time per pixel. Connectivity was defined as the number of cell pairs with a correlation coefficient larger than the cut-off value divided by the total number of the pairs of cells. This provided a measure of the degree of connections within a network. Small-world parameter, mean shortest path length ($\lambda$) and mean clustering coefficient ($\sigma$) were calculated by using the MATLAB BGL library (http://www.mathworks.com/matlabcentral/fileexchange/10922) and compared to corresponding randomized networks. Many biological networks have a small-world structure, where the mean shortest path length is as short as in random networks and the mean clustering coefficient is higher. This signifies that the average number of nodes (for example, neural cells) that a signal has to pass is low, and that many of the nodes are connected in clusters (*Watts and Strogatz, 1998*). A small-world network structure creates the possibilities of regional specialization and efficient signal transfer, and is a common organization of networks within the brain (*Telesford et al., 2011*).

Data were further processed to produce graphs in OriginPro, version 9.1 (OriginLab Corporation, Northamptom, MA, USA). Time-lapse $Ca^{2+}$ imaging time traces were normalized individually through $\Delta F/F_0$, where $\Delta F=F_1-F_0$. $F_1$ is the specific fluorescence intensity at a specific time point, and $F_0$ is the average intensity of 30 s before and after $F_1$.

A previously published toolbox was used for the frequency analysis of time traces (*Uhlén, 2004*).

Recordings of hypoglossal nerve activity were filtered (0.06-Hz low-pass), rectified and smoothed (1 s) (*Talpalar et al., 2011*) by using OriginPro (version 9.1, OriginLab Corporation, USA).

## Statistics

Statistical analysis of paired comparisons was performed by Student's t-test. Full factorial two-way ANOVA was performed when there was more than one independent variable or multiple observations. Both tests were two-sided. The compared data was of equal variance and normally distributed. All calculations for the statistical tests were conducted with JMP (v 11.1., SAS Institute Inc.,

**Table 10.** Statistical details for presented figures.

| Figure | Test used | Exact p-value | Degrees of freedom&F/t/z/R value |
|---|---|---|---|
| 1d | Full factorial two-way ANOVA | 0.0084 | F=0.612, DFE=21 |
| 1e | Full factorial two-way ANOVA | 0.0365 | F=0.284, DFE=21 |
| 1f | Full factorial two-way ANOVA | 0.0157 | F=0.329, DFE=21 |
| 1g | Full factorial two-way ANOVA | 0.0017<br>0.018<br>0.007 | F=0.547<br>F=0.332<br>F=1.618<br>DFE=23 |
| 2 – S1d | Student's t-test | 0.35<br>0.45 | DF = 9, 19 |
| 4e | Student's t-test | 0.63<br>0.76<br>0.91 | DF=11, 12, 8 |
| 4f | Student's t-test | 0.13, 0.34, 0.68<br>0.21, 0.28, 0.86<br>0.76, 0.76, 1.00 | DF=11, 12, 8 |
| 4 – S2 | Student's t-test | 0.43, 0.34, 0.12<br>0.11, 0.57, 0.19 | DF=1916 |
| 4 – S2 | Paired t-test | 0.02, 0.02, 0.04<br>0.03, 0.02, 0.01 | DF=1916 |
| 5b | Student's t-test | 0.42, 0.51, 0.80 | DF=15 |
| 5c | Student's t-test | 0.50, 0.62, 0.98 | DF=15 |
| 5 – S2a | Paired t-test | 0.0029 | DF=6 |
| 5 – S2b | Paired t-test | 0.041 | DF=10 |
| 5 – S2d | Paired t-test | 0.03 | DF=4 |
| 6k | Paired t-test | 0.03125<br>0.00391<br>0.3125<br>0.625 | DF=23 |
| 6l | Paired t-test | 0.01563<br>0.00781<br>0.28125<br>0.25 | DF=23 |
| 6i | Paired t-test | 0.492<br>0.331<br>0.390<br>0.390 | DF=23 |
| 6m | Paired t-test | 0.457<br>0.124<br>0.567<br>0.143 | DF=23 |
| 6 – S1a | Paired t-test | 0.15625<br>0.46094<br>0.0625<br>0.625 | DF=23 |
| 6 – S1b | Paired t-test | 0.15625<br>0.74219<br>0.15625<br>0.8125 | DF=23 |
| 6 – S1c | Paired t-test | 0.1246<br>0.07813<br>0.3125<br>0.8125 | DF=23 |

*Table 10 continued on next page*

*Table 10 continued*

| Figure | Test used | Exact p-value | Degrees of freedom&F/t/z/R value |
|---|---|---|---|
| 7b | Paired t-test | 0.018<br>0.034<br>0.047<br>0.079<br>0.132<br>0.084<br>0.028<br>0.063<br>0.067<br>0.012<br>0.077<br>0.90 | DF= 4, 3, 7 |
| 7g | Paired t-test | 0.95<br>0.51 | DF=4, 3 |
| 7-S1 | Paired t-test | 0.51861<br>0.15558<br>0.69733<br>0.51508<br>0.36415<br>0.2433 | DF=16 |
| 9e | Student's t-test | 0.15, 0.51, 0.57<br>0.29, 0.061, 0.0081 | DF=11, 10, 7 |
| 9f | Student's t-test | 0.38, 0.51, 0.75<br>0.59, 0.66, 0.99<br>0.10, 0.10, 0.95 | DF=11, 10, 7 |
| 9g | Paired t-test | 0.43 | DF=3 |
| 9h | Paired t-test | 0.0334 | DF=11 |
| 9i | Paired t-test | 0.0141<br>0.0283<br>0.00475 | DF=6, 5, 3 |
| 9 – S1 | Student's t-test | 0.77, 0.51, 0.92<br>0.28, 0.07, 0.60 | DF=935 |
| 9 – S1 | Paired t-test | 0.04, 0.04, 0.01<br>0.03, 0.02, 0.01 | DF=935 |
| 9 – S2 | Paired t-test | 0.00507<br>0.5745<br>0.22731<br>0.68788<br>0.81018<br>0.66252 | DF=16 |
| 10c | Paired t-test | 0.53, 0.20, 0.61<br>0.009, 0.015, 0.041<br>0.023, 0.045, 0.035<br>0.01, 0.09, 0.14 | DF=11, 7, 7, 9 |
| 10 – S1a | Paired t-test | 0.153<br>0.0848<br>0.388 | DF=6 |
| 10 – S1b | Paired t-test | 0.59<br>0.43 | DF=6 |
| 11b | Full factorial two-way ANOVA | 0.418<br>0.0161 | F=0.054<br>F=0.97 |
| 12b | Paired t-test | 0.00038<br>0.28 | DF= 6, 4 |
| 13b | Paired t-test | 0.13, 0.56 0.16<br>0.24, 0.12, 0.012 | DF=7 |
| 13d | Paired t-test | 0.00046<br>0.87 | DF=6, 4, 4, 3 |
| 13g | Paired t-test | 0.4112<br>0.0001 | DF=40 |

*Table 10 continued on next page*

Table 10 continued

| Figure | Test used | Exact p-value | Degrees of freedom&F/t/z/R value |
|--------|-----------|---------------|----------------------------------|
| 13 – S1c | Paired t-test | 0.0125<br>0.00098 | DF=5 |

Cary, NC, US). In all cases, p<0.05 was considered statistically significant. Data are presented as means ± SD. All data sets were compared less than 20 times, which is why no statistical corrections were made. As these experiments were expended to provide new descriptive data, no explicit power analysis was performed. Instead sample sizes similar to previous publications with similar methods were used. Details on the statistics are presented in *Tables 9* and *10*.

## Acknowledgements

We thank Markus Kruusmägi, Josephine Forsberg, Ruth Detlofsson, Dorina Ujvari, David Lagman, Torkel Mattesson and Lars Björk for their technical assistance; John Hayes for ImageJ scripts and advice; Marie Carlén and Thomas Ringstedt for their advice and discussion concerning optogenetics; Anders Blomqvist and Unn Örtegren Kugelberg (Linköping University, Sweden) for providing *Ptger3*[-/-] mice; and Ed Boyden for providing halorhodopsin-57. This study was supported by the Swedish Research Council, the Stockholm County Council, the Karolinska Institutet, and grants from the VINNOVA, M & M Wallenberg, Fraenkel, Axel Tielman, Freemasons Children's House and Swedish National Heart and Lung Foundations.

## Additional information

### Competing interests

EH: employed at the Karolinska University Hospital and the Karolinska Institutet and is a coinventor of a patent application regarding biomarkers and their relation to breathing disorders, WO2009063226. The other authors declare that no competing interests exist.

### Funding

| Funder | Grant reference number | Author |
|--------|------------------------|--------|
| Karolinska Institutet | Graduate MD PhD student fellowship | David Forsberg |
| Vetenskapsrådet | 2010-4392 | Per Uhlén |
| Hjärnfonden | FO2014+0220 | Per Uhlén |
| Vetenskapsrådet | 2013-3189 | Per Uhlén |
| Vetenskapsrådet | Senior Clinical researcher 6-year position, 2008-5829 | Eric Herlenius |
| VINNOVA | Future Health Innovation grant, 2010-00534 | Eric Herlenius |
| Stockholms Läns Landsting | 2011-0095 | Eric Herlenius |
| Hjärnfonden | FO2011-008 | Eric Herlenius |
| The Swedish National Heart and Lung foundation | 20120373 | Eric Herlenius |
| Vetenskapsrådet | 2009-3724 | Eric Herlenius |
| Knut och Alice Wallenbergs Stiftelse | Senior clinical researcher award and research grant, 102179 | Eric Herlenius |
| Stockholms Läns Landsting | 2012-0465 | Eric Herlenius |
| Stockholms Läns Landsting | 2014-0011 | Eric Herlenius |

| Hjärnfonden | FO2012-0036 | Eric Herlenius |
|---|---|---|
| Hjärnfonden | FO2015-0020 | Eric Herlenius |
| The Swedish National Heart and Lung foundation | 20150558 | Eric Herlenius |
| the Freemasons' Children's House | 2015 | Eric Herlenius |
| the Axel Tielman Foundation | 2015-00220 | Eric Herlenius |
| The Fraenckel Foundation | FRAE0018 | Eric Herlenius |

The funders had no role in study design, data collection and interpretation, or the decision to submit the work for publication.

## Author contributions

DF, ZH, Conception and design, Acquisition of data, Analysis and interpretation of data, Drafting or revising the article; ET, GS, YS, Acquisition of data, Analysis and interpretation of data, Drafting or revising the article; ES, KK, PU, Analysis and interpretation of data, Drafting or revising the article; EH, Conception and design, Analysis and interpretation of data, Drafting or revising the article

## Author ORCIDs

David Forsberg, http://orcid.org/0000-0002-4719-2201
Erik Smedler, http://orcid.org/0000-0003-4609-3620
Gilad Silberberg, http://orcid.org/0000-0001-9964-505X
Yuri Shvarev, http://orcid.org/0000-0001-6622-1453
Kai Kaila, http://orcid.org/0000-0003-0668-5955
Per Uhlén, http://orcid.org/0000-0003-1446-1062
Eric Herlenius, http://orcid.org/0000-0002-6859-0620

## Ethics

Animal experimentation: The studies were performed in strict accordance with European Community Guidelines and protocols approved by the regional ethic committee (Permit numbers: N247/13, N265/14b & N185/15).

## Additional files

### Supplementary files

• Source code 1. Cross-correlation analysis.

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
