## [Decision Letter]

Thank you for submitting your work entitled "CO_2_ induced release of PGE2 by glia increase inspiration and sighs through breathing brainstem neural networks" for consideration by *eLife*. Your article has been favorably evaluated by a Senior editor and three reviewers, one of whom, Jan-Marino Ramirez, is a member of our Board of Reviewing Editors.

The reviewers have discussed the reviews with one another and the Reviewing Editor has drafted this decision to help you prepare a revised submission.

Summary:

In the study by Forsberg et al., the authors combine in vivo, ex vivo and novel in vitro approaches to address the role of gap junctions and potential involvement of astrocytic release of prostaglandin E2 (PGE2) in respiratory control. PGE2 released in response to CO_2_ acts on prostaglandin EP3 receptors located in two main structures involved in respiratory rhythm generation, modulation and CO_2_ sensitivity: the pre-Bötzinger complex (preBötC) and the parafacial respiratory group/retrotrapezoid nucleus (pFRG/RTN). The authors propose that this PGE2-EP3R pathway is an essential component of the ventilatory response to CO_2_ and a vital modulator of respiratory activity, including sighing. The present results build upon previous results by Herlenius et al., showing that IL-1β impairs breathing during infection by inducing PGE2 release in the vicinity of brainstem respiratory networks (Hoffsteter et al. 2007), that inflammation via PGE2 and EP3R alters the ventilatory response to CO_2_ in brainstem spinal cord preparations (Siljehav et al., 2014), and other recent works on the role of PGE2 in the regulation of breathing pattern, in particular sigh frequency (Koch et al., 2015).

In order to characterize the pathway from brainstem astrocytes to the central pattern generator, the authors conducted plethysmographic studies in EP3R-knockout mice, which showed that PGE2 increases sighing activity via EP3R signaling and that the genetic ablation of EP3R reduces the hypercapnic response in vivo. In particular, they extended these experiments through electrophysiological studies in slice cultures of the brainstem with a functioning pFRG/RTN and preBötC. Furthermore, connectivity analysis of preBötC and pFRG/RTN respiratory networks (a very original aspect of the paper) showed that the hypercapnic response is dependent on gap junctions. The authors also used Ca^2+^ imaging and optogenetic silencing of EP3R-expressing cells to assess the role of this pathway in the hypercapnic response of the pFRG/RTN.

The reviewers were very impressed by the numerous techniques and novel approaches presented in this study. However, the manuscript is very complex and dense, and it tries to convey several different messages that in part become mixed up. For example, the thread of the Introduction oscillates between sighs, the role of astrocytes, inflammation, and inflammation-related respiratory disorders. A similar, general, concern also applies to the presentation of the results.

Another general concern is that the manuscript is lacking clear and concise specific hypotheses. The clarity of the manuscript would be greatly improved by clearly stating the hypothesis for a given set of experiments, and thus the conclusions of such experiments. At present, it is sometimes difficult to know how a given experiment relates to the overall conclusions. A similar concern relates to the figures. The reader must jump back and forth between different figures (and supplementary figures) to understand how the results relate to the specific and overall hypotheses of the manuscript.

It would vastly improve clarity to separate out analyses and hypotheses in the preBötC and those in pFRG/RTN. It would then be clearer why symmetrical analyses are not completed on each region.

Thus, in general, it will be critical to substantially re-organize the paper and figures, and make it more hypothesis-driven to bring it up to its full potential. Remember that the manuscript needs to be understood by a general audience.

Essential revisions:

1) The demonstration that it is actually the astrocytes that mediate the hypercapnic response is a minor and weaker part of the study. The study focuses more on the role of gap junctions than glia cells, and there is no experiment that actually tests the direct involvement of glia, or "astrocytic release" as stated in the Abstract. The role of glia is primarily inferred by prior work, e.g. by Gourine. The reviewers have no principle concern with the conclusion, and this possibility should be discussed, but the title, Introduction and Abstract should not imply that this study is about the role of glia. Along the same lines, the authors should avoid strong statements such as "PGE2 is released from astrocytes through CO_2_ sensitive connexin hemichannels". This is good for a discussion, but included into the Introduction (last paragraph) suggests that this will actually be demonstrated in the paper. Even with regards to the gap junctions. the authors did not really test gap junctions. The toxins used are not entirely specific. This should be discussed.

Along the same lines: A major conclusion of the manuscript is that PGE2 and ATP are released from astrocytes to alter pFRG/RTN responses to proton-independent increases in CO_2_ (Figure 12 and Discussion, sixth paragraph). However, this was not directly shown in the experiments conducted. The authors demonstrate changes after blocking gap junctions and blocking P2Rs alters the response, but this does not mean that astrocytes are releasing either PGE2 or ATP. The present results in context with previously published work does suggest this might be a logical conclusion, but alternative interpretations should be discussed.

2) The authors present a wealth of data using multiple different experimental methods, including a "breathing" brainstem slice culture. Several issues related to these slices were raised and are listed as separate points, below. Without addressing these concerns and being specific about the age of the slices in culture (i.e. 1, 2, or 3 weeks of culturing), it is difficult to critically evaluate the results. For clarity of the results, the reviewers suggest that it would be useful separate out data from the preBötC cultured slices and pFRG/RTN cultured slices, since the authors end up testing slightly different questions between the two preparations. This would help delineate which slice preparation was used for particular results. You will find related comments listed below.

3) Development of a "breathing" brainstem slice culture is very interesting and could be highly valuable to the field. However, the authors leave out a number of critical aspects with regards to the preparation that make the results difficult to evaluate. Firstly, it is unclear why such a preparation is necessary or what advantages it has compared to the classic rhythmic slice preparation. Acute measurements in the rhythmic slice preparation or a pFRG/RTN slice preparation would be just as advantageous for such experiments. Further, the authors are not clear about how long the slice culture is "breathing". No traces are shown to demonstrate baseline activity after 1, 2, or 3 weeks for the reader to see how activity changes with longer time in culture, and thus how the presented results might change with time. The authors do present data that the connectivity and number of active cells does not change with time (though the connectivity diagrams do appear to change over time especially when evaluating the raw numbers), but it is not clear how this would be observed in "breathing" or rhythmic/oscillatory behaviour. For example, is there any rhythmic activity after 3 weeks in culture in either the pFRG/RTN slices or the preBötC slices?

Additionally, no images are presented showing NK1 or other staining in the organotypic slices after three weeks, so it is difficult to decipher how neurons or astrocytes in culture have changed during this period (morphology, position, etc.). Since the conclusions in the study are based largely on data from such preparations, these are important methodological aspects to address. And since information is lacking in most figures and data presentation regarding the details of the slice preparation from which the data was obtained (i.e. 1, 2, or 3 weeks in culture), the reviewer assumes that the data from all weeks were collapsed, but without additional information, it is not clear whether such collapsing of data is appropriate. The authors also switch back and forth between saying "organotypic slice cultures", cultured cells, or slices. It would aid the reader if the authors used one terminology. Multiple types of preparations are used in these investigations and in the present format, it is difficult to follow which preparations are being used where and why such a preparation was necessary.

4) Related to the comment (3): There is a general issue with regards to the characterization of rhythmic activity in the two types of organotypic slices. Some of the specific concerns raised by the reviewers are as follows: The authors never spell out, whether they have population rhythms, and what kind of characteristics they have – even though they clearly have rhythms. The authors talk about "clustered activity", "clusters of high correlated activity". What do they mean with this? Only in the last few sentences of the subsection “Physiological measurements of brainstem respiratory activity demonstrate functional and responsive networks” they state that the preBötC remains rhythmic. The authors show something on Figure 3, but it doesn't look like rhythmicity. Yet, in the fourth paragraph of the aforementioned subsection, the authors imply that they have population rhythmicity that seems to even drive XII activity. Please try to be more concrete. Did these slices generate rhythmicity? What kind of properties do they find: frequency, amplitude, and also regularity? Do the calcium oscillations correspond to the "clustered" neuronal activities in frequency?

In the third paragraph of the subsection “Physiological measurements of brainstem respiratory activity demonstrate functional and responsive networks” – do the authors record from pacemaker neurons? This is potentially all very interesting, and if these slices show rhythmicity, the authors need to show traces of recordings, rather than activity from individual or pairs of neurons.

Figure 5 e.g. is beautiful, but do the pFRG/RTN slices show similar properties? Why is this not shown e.g. in Figure 7? Figure 4 is interesting, but why not show an expanded trace of rhythmicity? This is critical information in order to be able to assess what type of activity was characterized in this study. Did the pre-Bötzinger complex show activity that resembled sighs? Figure 11 shows some rhythmicity, but this should be expanded to see the characteristics. Given that this is the first report for these slices, a careful characterization is important. The authors might want to consider a reorganization of the manuscript: They should first characterize the population phenomenon, then report the clustered activity, correlations between neurons. Indeed, streamlining the messages, and not mixing up information would be critical to make this paper readable for a general readership.

5) Another general issue: the pFRG/RTN has been implicated (a) in CO_2_ sensing and (b) the neuronal control of active expiration. However, it is not entirely clear how these two functions work together, and whether the same neurons are responsible for active expiration and CO_2_ sensing. Phox2B is a good marker for the sensing aspect, but the rhythmicity observed in the pFRG/RTN slice may relate to "active expiration", and not necessarily to CO_2_ sensing. Clearly, the slice will also contain CO_2_ sensing neurons. Both possibilities are interesting, and both could play a role. The excitatory effect on the pFRG/RTN could imply that PGE2 induces active expiration, a possibility that has not been discussed in this paper. Discussing the role of active expiration should be considered, and the authors should also discuss the general issue of "rhythm generating function" versus "CO_2_ sensing", and the fact that this may not be related to the same neurons, but that PGE2 may affect both.

6) The reviewers have some reservations about the authors' claim that the PGE2-EP3R pathway is indispensable for the ventilatory response to hypercapnia. Specifically, the experimental support for this claim is not convincing (mainly in Figure 8 and Figure 11). There are several problems in Figure 8 that may question whether the inhibition of gap junctions by 18-a-GA abolished the ventilatory response to CO_2_. On the one hand, Panel b (top trace) and c (left graph) seem to indicate that 18-α-GA abolishes the response to CO_2_. On the other hand, in Panel c, the CO_2_ responses are the same whether gap junctions are inhibited by 18-α-GA or not (GZA, i.e. the negative control for the gap junction inhibitors 18α-GA). In fact, control values (in air) in the middle and right graphs are very similar, and the values obtained for GZA+ CO_2_ and CO_2_+18-α-GA are also very similar. This apparent contradiction is reflected in the legend: it is first stated that "hypercarbic responses were abolished when gap junctions were inhibited by 18-α-GA", but later on, the legend states that "the hypercarbic response was attenuated but not completely reversed by 18-α-GA". Some details in this figure also need attention. DF/F0 in Panel b should not be expressed in "arbitrary units", as mentioned. It is in fact a dimensionless variable (there is nothing to add after "DF/F0" on the y-axis label). Also, the scales are different for the different traces in Panel b, which is misleading. The experimental basis of the abolition of the CO_2_ response in Figure 11 also presents some shortcomings. Panel a indicates a lack of response to CO_2_ after AH6809 treatment, but there is no experimental control showing the time course of frequency changes in the absence of AH6808. In Panel f, the top graph lacks a control (e. g. a return to normocapnia). As presented, it is unclear whether the response to CO_2_ was abolished, as implied by the authors, or whether the period of measurement was too short for the full CO_2_ response to appear. In the bottom graph, the light stimulus seems to stabilize the hypercapnic response, not to abolish it, which departs from the clear-cut return to baseline presented in Panel g, left graph. This raises a concern on the way data were averaged in Panel g.

Incidentally, the DF/F0 graph in Panel a contains a numerical scale on the y-axis, which should be the case for all similar graphs (with a larger font size).

Because of all these considerations it seems an exaggeration to conclude, that the PGE2-EP3R pathway is "a vital modulator of respiratory activity", if only because EP3R^-/-^ mice (although obese) survive until adulthood.

7) The authors do not address why using a proton-independent CO_2_ stimulus was important to their studies, nor how they expect their studies to be altered if a proton-dependent stimulus was used. Since the majority of studies investigating hypercapnic responses has used acidification/pH changes as a stimulus (including the papers referenced in this manuscript), the authors should address this discrepancy. The authors need to address the rationale for choosing to investigate proton-independent changes in CO_2_ instead of changes in pH/H^+^, as is commonly done in the field. While CO_2_ may be a direct activator of cells, most literature to date has tested changes in pH. Additionally, many of the papers referenced in the manuscript in support of using proton-independent increased CO_2_, actually used pH to evoke changes. For example, Gourine et al. used changes in pH to examine involvement of ATP-P2Rs, yet the authors use these previous studies as rationale to look at the effects of proton-independent increased CO_2_ on the pFRG/RTN (Results, subsection “The pFRG/RTN respiratory region exhibited correlated network activity and retained CO_2_ sensitivity”). This discrepancy and rationale for using such a stimulus is not addressed in the present manuscript. Do the authors think that the same results would be obtained if experiments were conducted by altering pH? Why are the results proton-independent responses important? These questions are not addressed in the manuscript and would greatly improve the clarity and impact of the results.

8) The reviewers raised several questions regarding the experimental controls. Were the controls wild-type littermates of mutant mice (obtained by the crossing of heterozygous mice), and if so, were the heterozygotes also analyzed, as suggested by the mention that pups were analyzed after their genotype had been determined? If not, were the controls simply obtained from a C57Bl/6J strain? Control littermates are highly preferable, to control for date of birth, maternal care, litter size etc. These factors, also influence the maturation of CO_2_ sensitivity. It is indicated in the Methods that the EP3R gene was selectively deleted in knockout mice, as described in Fabre et al., 2001, but here, the authors used a 129 background.

9) The analysis of sighs is central to this study, and a very clear description of how sighs are defined and determined is thus essential. This point is ambiguous in the manuscript. In Methods, sighs are defined as "a breath with >50% increase in inspired volume compared to eupneic breaths", but there is no indication of how these reference eupneic breaths were determined. In Figure 1, not all sighs (arrows) seem to fulfill these criteria. In the legend for Figure 1, sighs are defined differently, as "an inspiratory volume and respiratory cycle period increase followed by a 2-cycle pause". In fact, apnea does not occur systematically after a sigh and cannot be a defining characteristic. Besides, the trace in Figure 1 that illustrates the authors' definition does not display any post-sigh pause. In contrast, a defining characteristic of sighs is their biphasic pattern (see Toporikova et al., 2015 and the references quoted in her article). Without taking into account this biphasic pattern, above-average breaths may be confused with sighs, especially in a context of increased variability. There are also several minor concerns with Figure 1: sigh frequencies (Panel c) do not exemplify the mean values shown in Panel d; in Panels c, e and f, only relevant comparisons are necessary. In Figure 1—figure supplement 1, it is puzzling that there are 11 divisions between 100 and 200 on the y-axis. Also, with a 60% increase in Fr (compared to WT) and a 30% increase in VT (compared to WT), the EP3R^-/-^ mice only displayed a 20-30% increase in VE (compared to WT). Is this correct?

10) Figure 1 needs to be revised: Three comments:

A) It is important to include "Figure 1—figure supplement 1" into the "final Figure 1". The supplementary figure contains an important message that is not contained in the Figure 1.

B) The traces in Figure 1 are not very helpful, as they contain lots of "artifacts" or deflections that are not respiratory, and could be mistaken by a non-specialist with sighs. I am very familiar with such traces and can detect sighs – but the general reader isn't an expert. Either, find other examples, or just show the graphs, which would also make it easier to add the supplementary figure.

C) Indeed, given that the sigh plays such an important part of the paper, there should be a figure of a sigh, and how the authors characterize sighs. Again, for the general reader this will not be clear from the figures presented. This comment relates also to the general comment (9) on the sigh.

11) Figure 12, has a very concrete model. It looks nice and is making their point. But, I am not convinced that they actually showed the differential glia/neuron modulation in preBötC and RTN/pFRG.

[Editors' note: further revisions were requested prior to acceptance, as described below.]

Thank you for resubmitting your work entitled "CO_2_-evoked release of PGE2 modulates sighs and inspiration as demonstrated in brainstem organotypic culture" for further consideration at *eLife*. Your revised article has been favorably evaluated by a Senior editor, a Reviewing editor, and two reviewers.

The manuscript has been improved but there are some remaining issues that need to be addressed before acceptance, as outlined below:

I would like to encourage you to cite and discuss the following study:

Organotypic slice cultures containing the preBötzinger complex generate respiratory-like rhythms.

Phillips WS, Herly M, Del Negro CA, Rekling JC.

J Neurophysiol. 2016 Feb 1;115(2):1063-70. doi: 10.1152/jn.00904.2015. Epub 2015 Dec 9.

PMID: 26655824

The publication of the organotypic slice does not diminish the impact of your study, but it would be important to acknowledge their study and also relate the activity that they report to your findings.

---

## [Author Response]

*The reviewers were very impressed by the numerous techniques and novel approaches presented in this study. However, the manuscript is very complex and dense, and it tries to convey several different messages that in part become mixed up. For example, the thread of the Introduction oscillates between sighs, the role of astrocytes, inflammation, and inflammation-related respiratory disorders. A similar, general, concern also applies to the presentation of the results.*

[…]

*It would vastly improve clarity to separate out analyses and hypotheses in the preBötC and those in pFRG/RTN. It would then be clearer why symmetrical analyses are not completed on each region.*

Thus, in general, it will be critical to substantially re-organize the paper and figures, and make it more hypothesis-driven to bring it up to its full potential. Remember that the manuscript needs to be understood by a general audience.

We agree that the manuscript and the data are complex and the thread was difficult to follow in the original version. In accordance with your advice, we have now reorganized the manuscript and figures. The hypothesis for each set of experiments is now more clearly stated. We have also reorganized the results so that they are now divided into three themes. After the introduction to the main issues from the in vivo data, we first characterize the new organotypic brainstem slice culture system we have developed and use this to investigate the mechanisms behind the results seen in vivo. Since those experiments did not answer our main questions fully, we have now created a novel brainstem slice culture for the pFRG/RTN and characterized it as well. Finally, in the third part, we used the pFRG/RTN culture to investigate the role of PGE_2_ in chemosensitivity. Consequently, we have split some figures and rearranged them to follow the new structure of the results.

*Essential revisions:*

*1) The demonstration that it is actually the astrocytes that mediate the hypercapnic response is a minor and weaker part of the study. The study focuses more on the role of gap junctions than glia cells, and there is no experiment that actually tests the direct involvement of glia, or "astrocytic release" as stated in the Abstract. The role of glia is primarily inferred by prior work, e.g. by Gourine. The reviewers have no principle concern with the conclusion, and this possibility should be discussed, but the title, Introduction and Abstract should not imply that this study is about the role of glia. Along the same lines, the authors should avoid strong statements such as "PGE2 is released from astrocytes through CO_2_ sensitive connexin hemichannels". This is good for a discussion, but included into the Introduction (last paragraph) suggests that this will actually be demonstrated in the paper. Even with regards to the gap junctions. the authors did not really test gap junctions. The toxins used are not entirely specific. This should be discussed.*

Along the same lines: A major conclusion of the manuscript is that PGE2 and ATP are released from astrocytes to alter pFRG/RTN responses to proton-independent increases in CO_2_ (Figure 12 and Discussion, sixth paragraph). However, this was not directly shown in the experiments conducted. The authors demonstrate changes after blocking gap junctions and blocking P2Rs alters the response, but this does not mean that astrocytes are releasing either PGE2 or ATP. The present results in context with previously published work does suggest this might be a logical conclusion, but alternative interpretations should be discussed.

We agree with your statement that our data do not show the direct release of PGE_2_ (or ATP) from astrocytes. We have rephrased these conclusions throughout the manuscript and removed the statement that PGE_2_ is released by glia from the title, Abstract, and Introduction and have clarified that this is still a working hypothesis. In the fourteenth paragraph of the Discussion, we discuss the issue further and describe how astrocytes are known to release ATP, for example, and that it would be possible for them to release the similar small molecule PGE_2_ as well. Notably, we show that astrocytes lining the ventral border express mPGEs^-1^ (which produces PGE_2_). However, as you correctly point out, we have not shown directly that the PGE_2_ released during exposure to elevated CO_2_ originates from astrocytes. This is a remaining issue that needs to be resolved. Sources like endothelial cells and microglia should be investigated in future studies.

Further, we are aware of the problem with the gap junction inhibitors. However, they are commonly used for this purpose and we use both vehicle alone and structurally similar substances as controls. In the fourth paragraph of the Discussion, we discuss the limitations of gap junction pharmacological tools and that the results concerning gap junction function should be interpreted carefully.

We have also removed statements that would draw a line between proton-independent and -dependent mechanisms. In our experimental setup, extracellular pH remained stable during hypercapnic exposure but we have not controlled or monitored intracellular pH. Thus, our present experiments and data do not distinguish between the two alternatives.

2) The authors present a wealth of data using multiple different experimental methods, including a "breathing" brainstem slice culture. Several issues related to these slices were raised and are listed as separate points, below. Without addressing these concerns and being specific about the age of the slices in culture (i.e. 1, 2, or 3 weeks of culturing), it is difficult to critically evaluate the results. For clarity of the results, the reviewers suggest that it would be useful separate out data from the preBötC cultured slices and pFRG/RTN cultured slices, since the authors end up testing slightly different questions between the two preparations. This would help delineate which slice preparation was used for particular results. You will find related comments listed below.

Information about the age of the cultures during the experiment is very important, and we understand that this has raised concerns. Throughout the revised manuscript, we have specified how long the slices were cultivated before the experiment was performed, and we have also provided more detailed data on the behavior of the brainstem slice culture at different ages. This is presented in Figure 2—figure supplement 1, Figure 4—figure supplement 2, Figure 5—figure supplement 1, and Figure 9—figure supplement 1. In Figure 4—figure supplement 2 and Figure 9—figure supplement 1 we also display the calcium signaling frequency and variability at the different culture ages. In general, all the experiments in which we characterize the model, i.e., survival, immunohistochemical analysis of protein expression, calcium activity, network structure, and testing for the preservation of hypercarbic responses, used slices 1, 2, or 3 weeks of age. The more detailed experiments concerning inflammation and hypercarbic responses were done using 1-week-old cultures.

As we describe above, we have separated the preBötC from the pFRG/RTN data.

*3) Development of a "breathing" brainstem slice culture is very interesting and could be highly valuable to the field. However, the authors leave out a number of critical aspects with regards to the preparation that make the results difficult to evaluate. Firstly, it is unclear why such a preparation is necessary or what advantages it has compared to the classic rhythmic slice preparation. Acute measurements in the rhythmic slice preparation or a pFRG/RTN slice preparation would be just as advantageous for such experiments. Further, the authors are not clear about how long the slice culture is "breathing". No traces are shown to demonstrate baseline activity after 1, 2, or 3 weeks for the reader to see how activity changes with longer time in culture, and thus how the presented results might change with time. The authors do present data that the connectivity and number of active cells does not change with time (though the connectivity diagrams do appear to change over time especially when evaluating the raw numbers), but it is not clear how this would be observed in "breathing" or rhythmic/oscillatory behaviour. For example, is there any rhythmic activity after 3 weeks in culture in either the pFRG/RTN slices or the preBötC slices?*

Additionally, no images are presented showing NK1 or other staining in the organotypic slices after three weeks, so it is difficult to decipher how neurons or astrocytes in culture have changed during this period (morphology, position, etc.). Since the conclusions in the study are based largely on data from such preparations, these are important methodological aspects to address. And since information is lacking in most figures and data presentation regarding the details of the slice preparation from which the data was obtained (i.e. 1, 2, or 3 weeks in culture), the reviewer assumes that the data from all weeks were collapsed, but without additional information, it is not clear whether such collapsing of data is appropriate. The authors also switch back and forth between saying "organotypic slice cultures", cultured cells, or slices. It would aid the reader if the authors used one terminology. Multiple types of preparations are used in these investigations and in the present format, it is difficult to follow which preparations are being used where and why such a preparation was necessary.

We are happy to hear that the development of a breathing brainstem organotypic culture might be highly valuable to the present field of research. In the short introduction to the first theme (Results), we point out that there is a need for a model system that provides the ability to perform long-term studies on respiratory centers. In the second paragraph of the Discussion, we now provide a more in-depth discussion on the pros and cons of our brainstem slice culture. In short, the culture provides the possibility of studying long-term drug exposure (hours to days), which we are currently conducting in parallel in studies that have not yet been finalized, and viral transduction in vitro (not possible in “acute slices”), which we have assessed for the optogenetic experiments in this study. Further, the slice cultures give the user somewhat more flexibility and significantly reduces the number of procedures otherwise needed to be performed on live animals.

To better characterize and clarify how the baseline activity of the cultures changes over time, we have added several figures and data. In Figure 4—figure supplement 2 and Figure 9—figure supplement 1 we have added traces of NK1R-expressing neurons’ activity during TTX inhibition at 7, 14, and 21 days in vitro. We have not detected differences among slices of different ages with any of the parameters we have analyzed. In Figure 5—figure supplement 1, we show rhythmic hypoglossal nerve output that is stable for over 2 hours of recording after 3 weeks of cultivation.

We have now shown the expression of the different neural markers after 1, 2, and 3 weeks of slice cultivation in Figure 2—figure supplement 2. Throughout the whole manuscript, we have better described the settings for each experiment, including cultivation stage. For consistency, we have used “brainstem slice culture” for general terminology.

The current characterization better presents the novel model system’s behavior over time. The present manuscript already contains a wealth of novel data, techniques, and approaches, as mentioned by the reviewers. Nonetheless, an even more detailed characterization will surely emerge in future studies. We hope that the novel approaches presented in the current manuscript will help to shed light on several of the remaining black boxes of how inspiration is generated and controlled.

*4) Related to the comment (3): There is a general issue with regards to the characterization of rhythmic activity in the two types of organotypic slices. Some of the specific concerns raised by the reviewers are as follows: The authors never spell out, whether they have population rhythms, and what kind of characteristics they have – even though they clearly have rhythms. The authors talk about "clustered activity", "clusters of high correlated activity". What do they mean with this? Only in the last few sentences of the subsection “Physiological measurements of brainstem respiratory activity demonstrate functional and responsive networks” they state that the preBötC remains rhythmic. The authors show something on Figure 3, but it doesn't look like rhythmicity. Yet, in the fourth paragraph of the aforementioned subsection, the authors imply that they have population rhythmicity that seems to even drive XII activity. Please try to be more concrete. Did these slices generate rhythmicity? What kind of properties do they find: frequency, amplitude, and also regularity? Do the calcium oscillations correspond to the "clustered" neuronal activities in frequency?*

*In the third paragraph of the subsection “Physiological measurements of brainstem respiratory activity demonstrate functional and responsive networks” – do the authors record from pacemaker neurons? This is potentially all very interesting, and if these slices show rhythmicity, the authors need to show traces of recordings, rather than activity from individual or pairs of neurons.*

Figure 5 e.g. is beautiful, but do the pFRG/RTN slices show similar properties? Why is this not shown e.g. in Figure 7? Figure 4 is interesting, but why not show an expanded trace of rhythmicity? This is critical information in order to be able to assess what type of activity was characterized in this study. Did the pre-Bötzinger complex show activity that resembled sighs? Figure 11 shows some rhythmicity, but this should be expanded to see the characteristics. Given that this is the first report for these slices, a careful characterization is important. The authors might want to consider a reorganization of the manuscript: They should first characterize the population phenomenon, then report the clustered activity, correlations between neurons. Indeed, streamlining the messages, and not mixing up information would be critical to make this paper readable for a general readership.

We hope our revised version better addresses your concerns on rhythmicity. We have also added more data to provide a more detailed description of the rhythmic activity (see above). The slices do indeed generate rhythmicity. Our results are based on calcium oscillations within the preBötC and extracellular measurements of the electrical hypoglossal nerve and nucleus activity. Results indicate the presence of NK1R^+^ neuronal pacemaker activity. Their rhythmic Ca^2+^ signals during inhibition of synaptic transmission remained, however with a lower frequency and higher coefficient of variation (subsection “Physiological measurements of brainstem respiratory activity demonstrate functional and responsive networks”, fifth paragraph). We have also added a display of a longer extracellular recording from the hypoglossal nucleus, showing activity after 3 weeks, as well as graphs that show frequency and regularity at 1, 2, and 3 weeks (Figure 5 andFigure 5—figure supplement 1).

Considering the clustered activity, we refer to cells that were shown to be connected in the correlation analysis. We have added Figure 4—figure supplement 1, for clarification and to make this concept easier to grasp. Also, calcium oscillation frequency does indeed correspond to motor output frequency.

Regarding sigh activity, we have noticed increases in the calcium signal amplitude and length after application of PGE_2_. This is discussed in the second paragraph of the subsection “PGE2 modulates preBötC activity”. We have not, however, identified sigh-related calcium activity in preBötC control recordings.

Concerning the pFRG/RTN output, to our knowledge, there is no established method to measure such electrical nerve output from slices, and hence such a technique is not established in our lab. Respiration-related output has been reported to emerge through the facial nerve. However, in those studies, preBötC and other respiratory centers were preserved in the specimen examined.

5) Another general issue: the pFRG/RTN has been implicated (a) in CO_2_ sensing and (b) the neuronal control of active expiration. However, it is not entirely clear how these two functions work together, and whether the same neurons are responsible for active expiration and CO_2_ sensing. Phox2B is a good marker for the sensing aspect, but the rhythmicity observed in the pFRG/RTN slice may relate to "active expiration", and not necessarily to CO_2_ sensing. Clearly, the slice will also contain CO_2_ sensing neurons. Both possibilities are interesting, and both could play a role. The excitatory effect on the pFRG/RTN could imply that PGE2 induces active expiration, a possibility that has not been discussed in this paper. Discussing the role of active expiration should be considered, and the authors should also discuss the general issue of "rhythm generating function" versus "CO_2_ sensing", and the fact that this may not be related to the same neurons, but that PGE2 may affect both.

Active expiration has not been the primary focus of this manuscript, but we have added the concept in the eleventh paragraph of the Discussion. It is possible that PGE_2_ stimulates both chemosensitive neurons and neurons important for active expiration. Such neurons could also overlap. Future studies should aim to investigate whether PGE_2_ also affects active expiration.

*6) The reviewers have some reservations about the authors' claim that the PGE2-EP3R pathway is indispensable for the ventilatory response to hypercapnia. Specifically, the experimental support for this claim is not convincing (mainly in Figure 8 and Figure 11). There are several problems in Figure 8 that may question whether the inhibition of gap junctions by 18-a-GA abolished the ventilatory response to CO_2_. On the one hand, Panel b (top trace) and c (left graph) seem to indicate that 18-α-GA abolishes the response to CO_2_. On the other hand, in Panel c, the CO_2_ responses are the same whether gap junctions are inhibited by 18-α-GA or not (GZA, i.e. the negative control for the gap junction inhibitors 18α-GA). In fact, control values (in air) in the middle and right graphs are very similar, and the values obtained for GZA+ CO_2_ and CO_2_+18-α-GA are also very similar. This apparent contradiction is reflected in the legend: it is first stated that "hypercarbic responses were abolished when gap junctions were inhibited by 18-α-GA", but later on, the legend states that "the hypercarbic response was attenuated but not completely reversed by 18-α-GA". Some details in this figure also need attention. DF/F0 in Panel b should not be expressed in "arbitrary units", as mentioned. It is in fact a dimensionless variable (there is nothing to add after "DF/F0" on the y-axis label). Also, the scales are different for the different traces in Panel b, which is misleading. The experimental basis of the abolition of the CO_2_ response in Figure 11 also presents some shortcomings. Panel a indicates a lack of response to CO_2_ after AH6809 treatment, but there is no experimental control showing the time course of frequency changes in the absence of AH6808. In Panel f, the top graph lacks a control (e. g. a return to normocapnia). As presented, it is unclear whether the response to CO_2_ was abolished, as implied by the authors, or whether the period of measurement was too short for the full CO_2_ response to appear. In the bottom graph, the light stimulus seems to stabilize the hypercapnic response, not to abolish it, which departs from the clear-cut return to baseline presented in Panel g, left graph. This raises a concern on the way data were averaged in Panel g.*

Incidentally, the DF/F0 graph in Panel a contains a numerical scale on the y-axis, which should be the case for all similar graphs (with a larger font size). Because of all these considerations it seems an exaggeration to conclude, that the PGE2-EP3R pathway is "a vital modulator of respiratory activity", if only because EP3R^-/-^ mice (although obese) survive until adulthood.

We agree that the PGE2-EP3R pathway is not indispensable, but our results show that it is important, given the attenuated response in EP3R knockout mice. Therefore, we have replaced “indispensable” and “vital” with “important” in the text.

The comment on Figure 8 is an interesting point. Indeed, there is no significant difference between frequencies during treatment with CO_2_+18-α-GA and CO_2_+GZA. However, the dynamics are different. When gap junctions are inhibited by 18-α-GA, no CO_2_ response is seen. This is not the case with GZA, as CO_2_ increases signaling frequency during GZA treatment. On the other hand, 18-α-GA is not able to completely reverse the CO_2_ response, supporting our theory that it is mainly pre-made PGE_2_ that is released. However, the reduction that 18-α-GA causes, indicates that gap junctions are still important to maintain the CO_2_ response, even if they are not exclusively responsible for this function. GZA is not able to reduce an already-initiated response. We have adjusted this in the figure (which is now Figure 10) to better illustrate this.

The DF/F0 panels have been adjusted throughout the manuscript according to your suggestion. We have also added the numerical scale to all relevant figures. We cannot, however, see how the scales are different in Figure 8 (now Figure 10). The middle (GZA) trace is somewhat noisier compared to the other two. Is this what you are referring to?

A control recording has been added in Figure 13 for comparison, and a graph showing average frequencies in these experiments has been added to Figure 13. A trace showing the return to normocapnia has been added to Figure 13, as you have suggested. Also, a cell trace (from the same experimental setting) that better represents the average data has been used for the bottom graph in Figure 13.

7) The authors do not address why using a proton-independent CO_2_ stimulus was important to their studies, nor how they expect their studies to be altered if a proton-dependent stimulus was used. Since the majority of studies investigating hypercapnic responses has used acidification/pH changes as a stimulus (including the papers referenced in this manuscript), the authors should address this discrepancy. The authors need to address the rationale for choosing to investigate proton-independent changes in CO_2_ instead of changes in pH/H^+^, as is commonly done in the field. While CO_2_ may be a direct activator of cells, most literature to date has tested changes in pH. Additionally, many of the papers referenced in the manuscript in support of using proton-independent increased CO_2_, actually used pH to evoke changes. For example, Gourine et al. used changes in pH to examine involvement of ATP-P2Rs, yet the authors use these previous studies as rationale to look at the effects of proton-independent increased CO_2_ on the pFRG/RTN (Results, subsection “The pFRG/RTN respiratory region exhibited correlated network activity and retained CO_2_ sensitivity”). This discrepancy and rationale for using such a stimulus is not addressed in the present manuscript. Do the authors think that the same results would be obtained if experiments were conducted by altering pH? Why are the results proton-independent responses important? These questions are not addressed in the manuscript and would greatly improve the clarity and impact of the results.

As stated above, our experiments do not distinguish between proton-dependent and -independent CO_2_ responses. With regard to intracellular pH, our experimental designs will have an effect e.g. (Ruusuvuori, Kirilkin et al. 2010).

8) The reviewers raised several questions regarding the experimental controls. Were the controls wild-type littermates of mutant mice (obtained by the crossing of heterozygous mice), and if so, were the heterozygotes also analyzed, as suggested by the mention that pups were analyzed after their genotype had been determined? If not, were the controls simply obtained from a C57Bl/6J strain? Control littermates are highly preferable, to control for date of birth, maternal care, litter size etc. These factors, also influence the maturation of CO_2_ sensitivity. It is indicated in the Methods that the EP3R gene was selectively deleted in knockout mice, as described in Fabre et al., 2001, but here, the authors used a 129 background.

Unfortunately, we do not have the ability to use littermate controls, as the strain is the homozygotic EP3R knockout obtained from the Anders Blomqvist research group at Linköping University, Sweden. Genotyping was performed to confirm the genetic homogeneity of the knockout line. These mice do indeed have a C57Bl/6J background and are regularly backcrossed with C57Bl/6J, which is why such animals, bred in the same facility, were used as wild-type controls. The reference by Fabre et al. simply describes the technique used, though in a different strain from the one used here. A clarification has been added to Methods, subsection “Subjects”, first paragraph.

9) The analysis of sighs is central to this study, and a very clear description of how sighs are defined and determined is thus essential. This point is ambiguous in the manuscript. In Methods, sighs are defined as "a breath with >50% increase in inspired volume compared to eupneic breaths", but there is no indication of how these reference eupneic breaths were determined. In Figure 1, not all sighs (arrows) seem to fulfill these criteria. In the legend for Figure 1, sighs are defined differently, as "an inspiratory volume and respiratory cycle period increase followed by a 2-cycle pause". In fact, apnea does not occur systematically after a sigh and cannot be a defining characteristic. Besides, the trace in Figure 1 that illustrates the authors' definition does not display any post-sigh pause. In contrast, a defining characteristic of sighs is their biphasic pattern (see Toporikova et al., 2015 and the references quoted in her article). Without taking into account this biphasic pattern, above-average breaths may be confused with sighs, especially in a context of increased variability. There are also several minor concerns with Figure 1: sigh frequencies (Panel c) do not exemplify the mean values shown in Panel d; in Panels c, e and f, only relevant comparisons are necessary. In Figure 1—figure supplement 1, it is puzzling that there are 11 divisions between 100 and 200 on the y-axis. Also, with a 60% increase in Fr (compared to WT) and a 30% increase in VT (compared to WT), the EP3R^-/-^ mice only displayed a 20-30% increase in VE (compared to WT). Is this correct?

We have better defined the sigh (as suggested in Toporikova 2015) on page 4 and in the Materials and methods section. In Figure 1, we now more illustratively show its characteristics. In Figure 1, we selected a different period from the same experimental setting that better represents the mean values presented in Figure 1. In Figure 1, there was a shift in the graph, but, after rendering it again, this has been solved. The differences in the figure supplement (which is now incorporated into the main Figure 1) as you observantly noted was due to a mix-up in the graphs. The Fq graph was the VE graph and vice versa. This embarrassing mistake has been corrected, and we sincerely apologize for this. The statements in the Results text, Table 1, and the legend were correct. The traces in the previous Figure 1 have been removed in accordance with your suggestion.

*10) Figure 1 needs to be revised: Three comments:*

*A) It is important to include "Figure 1—figure supplement 1" into the "final Figure 1". The supplementary figure contains an important message that is not contained in the Figure 1.*

*B) The traces in Figure 1 are not very helpful, as they contain lots of "artifacts" or deflections that are not respiratory, and could be mistaken by a non-specialist with sighs. I am very familiar with such traces and can detect sighs – but the general reader isn't an expert. Either, find other examples, or just show the graphs, which would also make it easier to add the supplementary figure.*

C) Indeed, given that the sigh plays such an important part of the paper, there should be a figure of a sigh, and how the authors characterize sighs. Again, for the general reader this will not be clear from the figures presented. This comment relates also to the general comment (9) on the sigh.

Please see the answer to comment (9).

*11) Figure 12, has a very concrete model. It looks nice and is making their point. But, I am not convinced that they actually showed the differential glia/neuron modulation in preBötC and RTN/pFRG.*

Thank you. The current Figure is now 14. We agree that we have not yet shown directly the differential glia/neuron modulation and have therefore added a question mark at the astrocyte to better illustrate that this is a possible mechanism that remains to be addressed in future work.

[Editors' note: further revisions were requested prior to acceptance, as described below.]

*The manuscript has been improved but there are some remaining issues that need to be addressed before acceptance, as outlined below:*

*I would like to encourage you to cite and discuss the following study:*

Organotypic slice cultures containing the preBötzinger complex generate respiratory-like rhythms.

Phillips WS, Herly M, Del Negro CA, Rekling JC.

J Neurophysiol. 2016 Feb 1;115(2):1063-70. doi: 10.1152/jn.00904.2015. Epub 2015 Dec 9.

*PMID: 26655824*

The publication of the organotypic slice does not diminish the impact of your study, but it would be important to acknowledge their study and also relate the activity that they report to your findings.

We have thoroughly read this study and find it interesting and intriguing as it both confirms and complements our data. We refer to it in the second paragraph of the Discussion.